# A Strong Baseline for Molecular Few-Shot Learning

**Philippe Formont**
*International Laboratory on Learning Systems, Montreal, Canada*
*Université Paris-Saclay, Ecole de technologie superieur, MILA*
**Hugo Jeannin**
*International Laboratory on Learning Systems, Montreal, Canada*
*Ecole de technologie superieur*
**Pablo Piantanida**
*International Laboratory on Learning Systems, Montreal, Canada*
*Université Paris-Saclay, MILA*
**Ismail Ben Ayed**
*International Laboratory on Learning Systems, Montreal, Canada*
*Ecole de technologie superieur*

**Reviewed on OpenReview:** *https: // openreview. net/ forum? id= JQOagisXny*

## Abstract

Few-shot learning has recently attracted significant interest in drug discovery, with a recent, fast-growing literature mostly involving convoluted meta-learning strategies. We revisit the more straightforward fine-tuning approach for molecular data, and propose a regularized quadratic-probe loss based on the the Mahalanobis distance. We design a dedicated block-coordinate descent optimizer, which avoid the degenerate solutions of our loss. Interestingly, our simple fine-tuning approach achieves highly competitive performances in comparison to state-of-the-art methods, while being applicable to black-box settings and removing the need for specific episodic pre-training strategies. Furthermore, we introduce a new benchmark to assess the robustness of the competing methods to domain shifts. In this setting, our fine-tuning baseline obtains consistently better results than meta-learning methods.

## 1 Introduction

Drug discovery is a process that aims to answer the following question: "How do we design molecules with biological activity on a specific therapeutic target and other necessary properties to become a drug?". To identify such compounds, scientists rely on a combination of experimental and computational methods (Xu & Hagler, 2002; Eckert & Bajorath, 2007). This lengthy and costly process takes on average 12 years and 1.8 billion dollars to bring a new drug to the market (Pammolli et al., 2019; Arrowsmith, 2011). To reduce the cost associated with this process and to speed up the development of new drugs, machine learning methods are often used to guide the discovery of new compounds (Gilmer et al., 2017; Chen et al., 2018). A common task for these models is to predict the activity of a compound on a given target exploiting the compound's structural or physio-chemical properties, referred to as quantitative structure-activity relationship (QSAR) (Tropsha, 2010). QSAR models can also be used as oracles to guide generative algorithms into generating compounds while optimizing their properties (Bengio et al., 2021; Blaschke et al., 2020; Gao et al., 2022).

**Low data regime.** The data exploited to train those models is scarce and costly to generate, as it results from numerous wet lab experiments. In practice, the size of the datasets corresponding to a novel target fluctuates along each project. Often, only up to a few hundred molecules are available to train the models. For this reason, various approaches for few-shot-learning in drug discovery have been proposed, attempting to provide reliable predictions while using a small amount of labelled data, where meta-learning seems to appear as the default and best solution (Schimunek et al., 2023; Chen et al., 2023; Altae-Tran et al., 2017).

**Meta-learning.** Meta-learning corresponds to a family of methods pre-trained to 'learn to learn' from a small set of labelled examples (support set), directly conditioning their predictions on this support set (Vinyals et al., 2016). Hence, these algorithms must pretrain a model specific to few-shot learning tasks. This implies that different pre-trained models must be maintained to perform few-shot and standard classification. Furthermore, most meta-learning algorithms use specific training procedures, as the training and test-time scenarios must be similar. Controlling the training procedure of these methods is hence necessary to use them. This can be problematic in a black-box setting, where only the predictions of the model can be queried through an API, and the model's weights cannot be accessed (Ji et al., 2021; Ouali et al., 2023). Using these methods in a black-box scenario would require imposing the training procedure on the server, which is not always possible, or would limit the number of models that could be used. Yet this black-box setting is especially valuable in drug discovery, where data is highly sensitive, and sharing a dataset or a pre-trained model on a private dataset is not always possible (Heyndrickx et al., 2024; Pejo et al., 2022; Smajić et al., 2023).

**Competitive fine-tuning methods.** Fine-tuning methods involve using a model pre-trained in a standard supervised setting and then fine-tuning it for few-shot classification. Unlike meta-learning, fine-tuning approaches utilize pre-trained models that can be applied to other tasks in standard machine learning, beyond few-shot classification, hence, a wider range of models can be used. In other domains, such as computer vision, meta-learning methods were shown to obtain only marginal improvements over more straightforward fine-tuning methods, using models pre-trained in a standard supervised setting (Chen et al., 2019; Luo et al., 2023). Some of these results were made possible thanks to the availability of foundational models pre-trained on large datasets available in these fields (Radford et al., 2021; Brown et al., 2020). However, developing such foundational models is a non-trivial task in the context of drug discovery, and is still an active research field (Seidl et al., 2023; Ahmad et al., 2022; Méndez-Lucio et al., 2022). In drug discovery, it was observed that **fine-tuning approaches yielded subpar performances** (Stanley et al., 2021), a finding seemingly at odds with existing results in vision applications.

**Contributions.** We re-visit fine-tuning black-box models using a simple Multitask backbone to perform QSAR few-shot learning. Our contributions could be summarized as follows:

- We evaluate the standard linear probe, a baseline often *overlooked in the molecular few-shot learning literature.*

- We generalize this standard cross-entropy baseline to a more expressive quadratic-probe loss based on the Mahalanobis distance and the class covariance matrices. We examine our loss from an optimization perspective, observe degenerate solutions and mitigate the issue with a block-coordinate descent and a dedicated regularization, which yields closed-form solutions at each optimization step. Interestingly, our quadratic probe achieves highly competitive performances as the size of the support set increases.

- Finally, in addition to evaluations on the standard FS-mol benchmark, we built two out-of-domain evaluation setups to evaluate the models' robustness to domain shifts. Our fine-tuning baselines obtain consistently better results than meta-learning algorithms.

## 2 Related Work

**Notations.** Assume that we are given a base dataset $\mathbb{D}_{\text{base}} = \{\mathbb{D}_{\text{base}}^t\}_{t \in \mathbb{T}_{\text{base}}}$, where $\mathbb{T}_{\text{base}}$ is a set of tasks. For a task $t$, we have $\mathbb{D}_{\text{base}}^t = \{\boldsymbol{x}_i, y_i\}_{0 \leq i \leq N_{\text{base}}^t}$, where $\boldsymbol{x}_i$ is a molecule and $y_i \in \{0, 1\}$ denotes its label.

We then have access to a test dataset defined similarly: $\mathbb{D}_{\text{test}} = \{\mathbb{D}_{\text{test}}^t\}_{t \in \mathbb{T}_{\text{novel}}}$ where $\mathbb{T}_{\text{novel}}$ is the set of new tasks ($\mathbb{T}_{\text{novel}} \cap \mathbb{T}_{\text{base}} = \emptyset$). Finally, for a task $t$, $\mathbb{D}_{\text{test}}^t$ is divided into two sets: $\mathcal{S}_t, \mathcal{Q}_t$ ($\mathcal{S}_t \cap \mathcal{Q}_t = \emptyset$) defining a set of labelled data that we will use to adapt the model, i.e., *support set* $\{\boldsymbol{x}_i, y_i\}_{i \in \mathcal{S}_t}$, and an unlabelled set whose labels are to be predicted, i.e., *query set* $\{\boldsymbol{x}_i, y_i\}_{i \in \mathcal{Q}_t}$.

The goal of the few-shot algorithms we study is to leverage the supervision information from the support samples to predict the labels within the query set.

**Meta-learning.** Training a meta-learning model, referred to as meta-training, differs from a classical supervised procedure by the fact that the training set is divided into tasks. Each task is divided into a support set and a query set that will be fed into the model to perform its training. This data processing is specific to meta-learning.

The work of Stanley Stanley et al. (2021) introduced a benchmark for molecular few-shot learning, FS-mol, and evaluated various methods. For instance, the authors trained an adaptation of Protonet (Snell et al., 2017) with a graph neural network (GNN). Besides, the authors pre-trained a graph neural network (GNN) with the MAML algorithm (Finn et al., 2017), a model aiming at learning a good initialization of the model's weights, such that the model can be quickly adapted to new tasks. Building on the MAML paradigm, the property-aware relation network (PAR) (Wang et al., 2021) builds a similarity graph between the molecules of the support set and the queried molecule and extracts from this graph the molecule's predicted class. In recent work, Chen et al. (2023) proposed a method separating the parameters $\theta_{meta}$ that are updated during meta-training, while the other parameters $\theta_{\text{adapt}}$ are only updated during the adaptation. The back-propagation on each task is done by using Cauchy's implicit function theorem, which requires estimating the hessian of the loss $\frac{\partial^2 \mathcal{L}_{\text{support}}}{\partial \theta_{\text{adapt}}^2}$ on the support set. Finally, some work evaluated the effect of leveraging a large set of context molecules to construct similarity measures between the support set's molecules and the queried molecule using modern hopfield networks (Schimunek et al., 2023).

**Fine-tuning methods.** Among the baselines presented by Stanley *et al.* (Stanley et al., 2021), some of them used a backbone trained without relying on meta-training. First, the authors fine-tuned a GNN trained with a multitask objective. This model consists in a GNN common trunk followed by a 2-layer task-specific head. At test time a new task-specific head is added corresponding to the new task to adapt on. Both the common-trunk and the task-specific parameters are fine-tuned on the support set. A second approach used the molecular attention transformer (MAT) (Maziarka et al., 2019) and fine-tuned it on the support set of the novel task. **Both methods achieved performances on FS-mol below the aforementioned meta-learning methods.** Furthermore, they need to fine-tune the whole model, which can be computationally expensive and requires the user to access the model's weights at test time. Hence, these methods cannot be directly used in a black-box setting. In the recent years, a significant effort was made to build foundational models able to handle molecular data (Méndez-Lucio et al., 2022; Beaini et al., 2024; Khan et al., 2024; Xia et al., 2023), but very few proposed methods assess the ability of these models to perform few-shot classification. Notably, the work of Seidl et al. (2023) proposed a multi-modal model leveraging both the molecular graph and a textual representation of the assays to make predictions, enabling the creation of zero-shot models. The authors evaluated the model on FS-mol with a linear probe. However, to train their model, the support set of all tasks was integrated into the pre-training set, which breaks the assumption $\mathbb{T}_{novel} \cap \mathbb{T}_{base} = \emptyset$. This evaluation corresponds to an assessment of the model's ability to generalize on previously seen tasks with few labels, not the model's ability to adapt to unseen tasks.

## 3 Methods

### 3.1 Model pre-training

Various methods exist to pre-train models for drug discovery (Hu* et al., 2020), which can usually be divided into supervised and self-supervised methods. Supervised pre-training refers to methods training a model on a dataset of molecules relying on the association between the input and labels. In contrast, self-supervised pre-training refers to methods training a model on a dataset of molecules without using or knowing the labels. While self-supervised pre-training enables the use of larger data sets, supervised pre-training tends to be more effective when the labels used during training correspond to assays similar to those of the downstream task (Sun et al., 2022). In this work, we pursue the supervised setting via a multitask model based on the training set of the FS-mol benchmark (Stanley et al., 2021). We focus our work on QSAR modeling, evaluating the ability of the pre-training backbone to adapt to new tasks. An evaluation of pretrained models originating from the molecular representation learning literature is available in Appendix A.6

The GNN backbone $f_\theta$ will take the molecular graph and the fingerprints as input, based on a principal neighborhood aggregation model (Corso et al., 2020). We add a task-specific head $g_\phi$ to the backbone, a

one-layer binary classifier during the pre-training. Once the training is finished, we only keep the parameters $\theta$ of the backbone. The molecular embeddings passed to the few-shot classifier are obtained by

$$\boldsymbol{z}_i = \frac{f_\theta(\boldsymbol{x}_i)}{\|f_\theta(\boldsymbol{x}_i)\|} = \frac{f_\theta(\mathcal{G}_i, \boldsymbol{f}_i)}{\|f_\theta(\mathcal{G}_i, \boldsymbol{f}_i)\|},$$

with $\mathcal{G}_i$ denoting the molecular graph, $\boldsymbol{f}_i$ the molecular fingerprint.

## 3.2 Multitask Linear Probing

As explained in the section 2, multitask models were shown to be outperformed by most other methods (Stanley et al., 2021). Still, we show that we can obtain better classification capabilities by following the paradigm of linear probing (Chen et al., 2019). Linear probing corresponds to a model training the parameters $\boldsymbol{w}_k, \|\boldsymbol{w}_k\| = 1$ for each class $k \in C = \{0, \dots, n_{\text{class}}\}$. The predicted probability of a molecule $\boldsymbol{x}_i$ to belong to the class $k \in C$ is then given by:

$$p_{i,k} = \frac{\exp\left(\tau \langle \boldsymbol{z}_i, \boldsymbol{w}_k \rangle + b_k\right)}{\sum_{k' \in C} \exp\left(\tau \langle \boldsymbol{z}_i, \boldsymbol{w}_{k'} \rangle + b_{k'}\right)} \tag{1}$$

where $b_k$ is the model's bias, $\tau$ is a temperature hyperparameter, and for the clarity of the notation, we omitted the dependency on the task $\boldsymbol{t}$. The linear probe's parameters are then optimized to minimize the cross-entropy loss on the support set.

## 3.3 Quadratic probing

**Motivation.** We propose an extension of the aforementioned linear probe, which is more competitive as the support set grows. Instead of the cosine similarity used in the linear probe, we propose to use the Mahalanobis distance between the query points and the class prototypes:

$$\|\boldsymbol{z}_i - \boldsymbol{w}_k\|^2_{\Sigma_k^{-1}} = (\boldsymbol{z}_i - \boldsymbol{w}_k)^T \Sigma_k^{-1} (\boldsymbol{z}_i - \boldsymbol{w}_k), \tag{2}$$

where $\Sigma_k$ is a positive semi-definite (PSD) matrix. For simplicity, we will refer to $\mathbf{M}_k \triangleq \Sigma_k^{-1}$ as the precision matrix of the distribution to simplify the notations.

The predicted probability of a molecule $\boldsymbol{x}_i$ to belong to the class $k \in C$ can be then obtained as:

$$p_{i,k} \triangleq \frac{\exp\left(-\|\boldsymbol{z}_i - \boldsymbol{w}_k\|^2_{\mathbf{M}_k}\right)}{\sum_{k' \in C} \exp\left(-\|\boldsymbol{z}_i - \boldsymbol{w}_{k'}\|^2_{\mathbf{M}_k}\right)}. \tag{3}$$

The Mahalanobis distance measures the distance between a point and a normal distribution of mean $\boldsymbol{w}_k$ and covariance matrix $\Sigma_k$. This distance is more expressive than the cosine similarity, as it considers the covariance matrix of the class. In fact, one can see it as generalization of the linear probe,

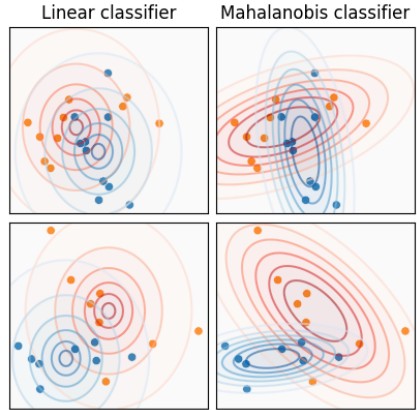

Figure 1: Values of the logits produced by two different classifiers: a linear probe (left) and a quadratic probe based on the Mahalanobis distance (right), in a two-dimensional feature space with two classes.

which corresponds to choosing a covariance matrix proportional to identity. Figure 1 depicts an illustration of the difference between linear probe and our quadratic probe.

In the next section, we show that directly minimizing the cross-entropy loss over $\mathbf{M}_k$ yields degenerate solutions, in which the Frobenius norms of the covariance matrices go to infinity. To mitigate this issue, we propose a surrogate objective, which corresponds approximately to augmenting the cross-entropy loss with a regularization term based on the Frobinous norms of the covariance matrices, while yielding closed-form solutions at each optimization step w.r.t $\mathbf{M}_k$. We proceed with block-coordinate descent approach to optimize our regularized quadratic-probe loss, updating the parameters $\{\boldsymbol{w}_k\}_{k \in \{0,1\}}$ and $\{\mathbf{M}_k\}_{k \in \{0,1\}}$ separately.

**Gradient descent.** Optimizing the parameters of the quadratic probe is a challenging task, as the estimator's number of parameters is much larger than the linear probe's. The first greedy option would be to optimize $\mathbf{M}_k$ by gradient descent on the support set. We propose to optimize $\mathbf{D}_k \triangleq \mathbf{M}_k^{-1/2}$ ($\mathbf{M}_k = \mathbf{D}_k^T \mathbf{D}_k = \Sigma_k^{-1}$) as a parameter of the model to ensure that $\mathbf{M}_k$ is PSD.

However, by following this procedure, the model has a high chance of over-fitting. In particular, if the classes are linearly separable, the model will find a minimum of the cross-entropy loss, pushing $\|\mathbf{M}_k\|$ to become arbitrarily large.

**Proposition 3.1.** *Let $\Theta = \{\boldsymbol{w}_k, \mathbf{M}_k\}_{k \in \{0,1\}}$, and we will note $p_\Theta(k|\boldsymbol{z}) = p_{i,k}$ as described in Equation 3 (to highlight the dependency to the parameters $\Theta$).*

*If the samples from both classes are linearly separable, we can construct a set of parameters, $\Theta(\lambda) = \{\boldsymbol{w}_k, \mathbf{M}_k(\lambda)\}_{k \in \{0,1\}}$ such that:*

$$\begin{cases} \forall i \in \{0, \dots, N\}, \quad \lim_{\lambda \to +\infty} \mathcal{L}_{ce}(\boldsymbol{z}_i, y_i, \Theta(\lambda)) = 0, \\ \forall k \in \{0,1\}, \quad \lim_{\lambda \to +\infty} \|\mathbf{M}_k(\lambda)\|_F = +\infty, \end{cases}$$

*where $\mathcal{L}_{ce}(\boldsymbol{z}_i, y_i, \Theta) = -\log p_\Theta(y_i|\boldsymbol{z}_i)$ is the point-wise cross-entropy loss w.r.t our model.*

The proof of proposition 3.1 is relegated to Appendix A.2. This proposition relies on the assumption that the data points are linearly separable. In the few-shot learning setting, this assumption is often verified. Indeed, as the data dimension is much larger than the number of data points, Cover's theorem (Cover, 1965) states that the points are almost certainly linearly separable. Another argument supporting the assumption of linear separability comes from an approximation often made in high-dimensional spaces: if $d \gg N$, $\quad \forall i, j \in \{0, \dots, N\}^2 \quad \boldsymbol{z}_i^t \boldsymbol{z}_j = 0$. Then, the vector $\boldsymbol{v} = \sum_{i=0}^N y_i \boldsymbol{z}_i - (1 - y_i)\boldsymbol{z}_i$ defines a hyperplane separating data points in both classes.

**Block coordinate optimization and modified loss.** Instead of optimizing the parameters of the quadratic probe by gradient descent, we optimize the parameters of the model with a block coordinate descent, updating the parameters $\{\boldsymbol{w}_k\}_{k \in \{0,1\}}$ and $\{\mathbf{M}_k\}_{k \in \{0,1\}}$ separately. In particular, we will perform one gradient descent step on $\{\boldsymbol{w}_k\}_{k \in \{0,1\}}$ using the cross-entropy loss and then minimize a modified loss w.r.t $\{\mathbf{M}_k\}_{k \in \{0,1\}}$, that does not suffer from the issue described in proposition 3.1. In the following section, $\{\boldsymbol{w}_k\}_{k \in \{0,1\}}$ is considered fixed.

Decomposing the cross-entropy in two parts, we first identify the term pushing the parameters $\mathbf{M}_k$ to be arbitrarily large. The cross-entropy loss of the quadratic probe can be divided into two terms:

$$\mathcal{L}_{ce}(\mathcal{S}, \Theta) = \frac{1}{|\mathcal{S}|} \sum_{i \in \mathcal{S}} \left( f_1(\boldsymbol{z}_i, y_i, \Theta) + f_2(\boldsymbol{z}_i, \Theta) \right), \tag{4}$$

where

$$f_1(\boldsymbol{z}_i, y_i, \Theta) \triangleq \|\boldsymbol{z}_i - \boldsymbol{w}_{y_i}\|_{\mathbf{M}_{y_i}}^2,$$

$$f_2(\boldsymbol{z}_i, \Theta) \triangleq \log \left( \sum_{k \in \{0,1\}} \exp\left(-\|\boldsymbol{z}_i - \boldsymbol{w}_k\|_{\mathbf{M}_k}^2\right) \right). \tag{5}$$

*Remark* 3.2. For all $i \in \mathcal{S}$, $f_1(\boldsymbol{z}_i, y_i, \Theta)$ is minimized when $\|\mathbf{M}_k\|_F \to 0$, and $f_2(\boldsymbol{z}_i, y_i, \Theta)$ as $\|\mathbf{M}_k\|_F \to +\infty$.

The function $f_2$ is hence responsible for the degenerate solutions obtained while minimizing the cross-entropy loss. We propose to replace $f_2$ with a different function $\tilde{f}_2$, playing the same role pushing $\|\mathbf{M}_k\|_F$ towards infinity, but with a less aggressive behaviour. The new loss obtained $f_1 + \tilde{f}_2$ admits a closed-form solution on $\mathbf{M}_k$, for a fixed $\boldsymbol{w}_k$, which corresponds to the estimator of the empirical covariance matrix's inverse. The expression of $\tilde{f}_2$ is then:

$$\tilde{f}_2(\Theta) \triangleq - \sum_{k \in \{0,1\}} \frac{|\mathcal{S}_k| \log \det(\mathbf{M}_k)}{|\mathcal{S}|},$$

where $\mathcal{S}_k = \{i \in \mathcal{S} | y_i = k\}$.

**Proposition 3.3.** *Both $f_2$ and $\tilde{f}_2$ are optimized when the norms of $\{\mathbf{M}_k\}_{k \in \{0,1\}}$ approach infinity. When for all $k \in \{0,1\}$, the eigenvalues of $\mathbf{M}_k$ grow to infinity:*

$$f_2\left(\boldsymbol{z}_i, \Theta\right) = O(\sum_{k \in \{0,1\}} \|\mathbf{M}_k\|_F),$$

$$and \quad \tilde{f}_2\left(\boldsymbol{z}_i, \Theta\right) = O(\sum_{k \in \{0,1\}} \log\|\mathbf{M}_k\|_F), \tag{6}$$

*And the resulting loss $f_1 + \tilde{f}_2$ admits a closed-form solution, for $k \in \{0,1\}$:*

$$\mathbf{M}_k = \left( \frac{1}{|\mathcal{S}_y|} \sum_{i \in \mathcal{S}_y} \left(\boldsymbol{z}_i - \boldsymbol{w}_k\right)\left(\boldsymbol{z}_i - \boldsymbol{w}_k\right)^T \right)^{-1}. \tag{7}$$

*Remark* 3.4. Minimizing $f_1 + \tilde{f}_2$ is equivalent to maximizing the log-likelihood of a multivariate Gaussian distribution with mean $\boldsymbol{w}_k$ and covariance matrix $\Sigma_k$, explaining why the closed form solution we obtain corresponds to the estimator of the empirical covariance matrix's inverse. Empirical results displaying the evolution of the largest eigenvalue of $\mathbf{M}_k$ using gradient descent and the quadratic probe are displayed in Appendix A.3

The function $\tilde{f}_2$ can hence be seen as an upper bound on $f_2$ when $\|\mathbf{M}_k\|_F$ becomes large, for $k = 0, 1$. Indeed, $\tilde{f}_2$ exerts less pressure on expanding $\mathbf{M}_k$ to larger matrices. This replacement can approximately be seen as adding the regularization terms $\alpha\|\mathbf{M}_k\|_F - \beta \log \|\mathbf{M}_k\|_F$ to the loss ($\alpha$ and $\beta$ are constants).

Finally, we train the quadratic probe by updating $\mathbf{M}_k$ using the closed-form solution obtained above (Equation 7 and performing a gradient step on $\boldsymbol{w}_k$ using the cross-entropy loss. Since the estimation of the covariance matrix can be unstable, we use a shrinkage parameter $\lambda \in [0, 1]$ to stabilize the estimation of the covariance matrix ($\Sigma_k \triangleq (1 - \lambda)\Sigma_{k,\text{empirical}} + \lambda\mathbf{I}$). We consider $\lambda$ as a hyperparameter of the model to be optimized (refer to Appendix A.4 for further details). The algorithm of the training process of the quadratic probe can be found in Appendix A.1.

## 4 Experimental Results

### 4.1 FS-mol benchmark

**Dataset.** We evaluate our methods on the FS-mol benchmark (Stanley et al., 2021). This benchmark aims to classify molecules into active/inactive binary classes for a given therapeutic target, divided into 5000 tasks (40 in the validation and 157 in the test sets). The tasks the models are evaluated on are balanced, with on average 47% of active molecules, a value ranging from 30% to 50%.

**Methods compared.** We compare our methods to various meta-learning algorithms: GNN-MAML (Finn et al., 2017), Protonet (Snell et al., 2017), ADKT-IFT (Chen et al., 2023), the Property Aware Relation network (PAR) (Wang et al., 2021), and conditional neural process (CNP) (Garnelo et al., 2018), MHNFS (Schimunek et al., 2023).[1] We report the results of fine-tuning baselines such as fully fine-tuning of a molecular attention transformer (MAT) (Maziarka et al., 2019) and GNN-MT, their multitask fine-tuning approach. We also re-evaluated the few-shot adaptation capabilities of CLAMP (Seidl et al., 2023) on the FS-mol benchmark (without adding the support set of each task to the model's pre-training set). We also consider simple baselines: training a random forest classifier and a kNN on the support set's molecular fingerprint (Stanley et al., 2021), and Similarity Search, a standard chemoinformatic method relying only on the Tanimoto similarity between these fingerprints (Bender et al., 2004; Cereto-Massagué et al., 2015).. The authors of FS-mol also propose to train a single-task GNN on the support set of a task as a baseline(GNN-ST).

**Evaluation procedure.** To compare few-shot learning methods, all models were pre-trained on FS-mol's training set. Models are evaluated using the improvement of the average precision compared to a random

---

[1]We used the author's scripts to train the model while implementing the evaluation scripts, which were unavailable on the author's repository.

Table 1: Results on FS-mol ($\Delta AUCPR$) of each method for support set sizes of 16, 32, 64 and 128 (averaged over 10 runs). The best models, or models whose average performances are in the confidence interval of the best model, are highlighted in bold.

| MODEL-TYPE | MODEL | $|\mathcal{S}| = 16$ | $|\mathcal{S}| = 32$ | $|\mathcal{S}| = 64$ | $|\mathcal{S}| = 128$ |
|---|---|---|---|---|---|
| BASELINES | KNN | $0.051_{\pm 0.005}$ | $0.076_{\pm 0.006}$ | $0.102_{\pm 0.007}$ | $0.139_{\pm 0.007}$ |
| | GNN-ST | $0.021_{\pm 0.005}$ | $0.027_{\pm 0.005}$ | $0.031_{\pm 0.005}$ | $0.052_{\pm 0.006}$ |
| | RF | $0.093_{\pm 0.007}$ | $0.125_{\pm 0.008}$ | $0.163_{\pm 0.009}$ | $0.213_{\pm 0.009}$ |
| | SIMSEARCH | $0.113_{\pm 0.009}$ | $0.142_{\pm 0.008}$ | $0.169_{\pm 0.008}$ | $0.210_{\pm 0.008}$ |
| META-LEARNING | PAR NeurIPS2021 | $0.130_{\pm 0.009}$ | $0.140_{\pm 0.009}$ | $0.147_{\pm 0.009}$ | $0.161_{\pm 0.009}$ |
| | GNN-MAML | $0.160_{\pm 0.009}$ | $0.167_{\pm 0.009}$ | $0.174_{\pm 0.009}$ | $0.192_{\pm 0.009}$ |
| | CNP | $0.187_{\pm 0.011}$ | $0.192_{\pm 0.011}$ | $0.195_{\pm 0.011}$ | $0.208_{\pm 0.011}$ |
| | PROTONET NeurIPS2021 | $0.206_{\pm 0.009}$ | $0.242_{\pm 0.009}$ | $0.271_{\pm 0.009}$ | $0.301_{\pm 0.009}$ |
| | MHNFS ICLR 2023 | $0.205_{\pm 0.010}$ | $0.215_{\pm 0.009}$ | $0.219_{\pm 0.009}$ | $0.231_{\pm 0.009}$ |
| | ADKF-IFT ICLR 2023 | $\mathbf{0.231_{\pm 0.009}}$ | $\mathbf{0.263_{\pm 0.010}}$ | $\mathbf{0.287_{\pm 0.010}}$ | $\mathbf{0.318_{\pm 0.010}}$ |
| FINE-TUNING | MAT | $0.052_{\pm 0.005}$ | $0.069_{\pm 0.006}$ | $0.092_{\pm 0.007}$ | $0.136_{\pm 0.009}$ |
| | GNN-MT | $0.112_{\pm 0.006}$ | $0.144_{\pm 0.007}$ | $0.177_{\pm 0.008}$ | $0.223_{\pm 0.009}$ |
| | CLAMP ICLR 2023 | $0.202_{\pm 0.009}$ | $0.228_{\pm 0.009}$ | $0.248_{\pm 0.009}$ | $0.271_{\pm 0.009}$ |
| | LINEAR PROBE | $\mathbf{0.224_{\pm 0.010}}$ | $0.252_{\pm 0.010}$ | $0.273_{\pm 0.010}$ | $0.301_{\pm 0.009}$ |
| | QUADRATIC PROBE | $\mathbf{0.227_{\pm 0.010}}$ | $\mathbf{0.255_{\pm 0.010}}$ | $\mathbf{0.279_{\pm 0.010}}$ | $\mathbf{0.310_{\pm 0.010}}$ |

classifier ($\Delta AUCPR$). Hyper-parameters are selected through a hyper-parameter search on the validation set across all support set sizes. All hyperparameters are kept constant over each support set size, except for the optimal numbers of epochs for our fine-tuning baselines, which are fitted for each support set size on the validation set. We present the results for support set sizes of 16, 32, 64 and 128 (larger support set sizes would discard some tasks from the test set).

**Results.** Table 1 presents the results of the different methods on the FS-mol test set. First, our evaluation of CLAMP achieves results similar to the ones reported in the original paper with 16 labelled points and obtains better results with 64 and 128 labelled points. On the other hand, we did not succeed in reproducing MHNFS's results with our implementation.

The performances of both the linear and quadratic probes outperform the other fine-tuning baselines. Using a basic pre-training procedure, our fine-tuning methods also provide strong baselines for the few-shot adaptation of pre-trained models, to estimate the relevance of a new pre-training procedure in this setup.

As expected, the quadratic probe obtains consistently slightly better results than the linear probe, and the difference becomes more significative as the amount of labeled data increases. This result is expected since the quadratic probe is a model fine-tuning more parameters than the linear probe, at the cost of a slightly more complex procedure (requiring to estimate $\mathbf{M}_k$). Additionally, the quadratic probe consistently reaches performances that are competitive with ADKT-IFT's results at each support set size.

### 4.2 Ablation on the Free optimisation of $\Sigma_k$

As explained in section 3, we optimize $\Sigma_k$ by taking the empirical covariance matrix of the embeddings, considering $\mathbf{w}_k$ is the mean of the distribution. Figure 2 highlights the difference between optimizing $\Sigma_k$ with

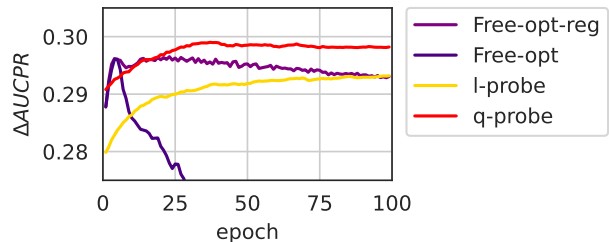

Figure 2: Evolution of the model's performance on the validation set during the few-shot adaptation. Free-opt refers to performing a gradient descent on $\mathbf{M}_k$, Free-opt-reg adds a regularisation on the norm of $\mathbf{M}_k$. ($|\mathcal{S}| = 64$)

Table 2: Results on DTI tasks ($\Delta AUCPR$) for support set sizes of 16 and on the library screening tasks (ranks).

| Model | KIBA | DAVIS $|\mathcal{S}| = 16$ | BindingDB_Kd | Library Screening | | | |
|---|---|---|---|---|---|---|---|
| | | | | $|\mathcal{S}| = 16$ | $|\mathcal{S}| = 32$ | $|\mathcal{S}| = 64$ | $|\mathcal{S}| = 128$ |
| Similarity search | $17.3_{\pm 0.9}$ | $19.1_{\pm 3.1}$ | $17.3_{\pm 1.3}$ | 3.97 | 3.69 | **3.19** | **2.78** |
| Protonet NeurIPS2021 | $25.1_{\pm 1.0}$ | $37.6_{\pm 1.6}$ | $34.0_{\pm 0.9}$ | 3.66 | 3.57 | 3.32 | 3.17 |
| ADKF-IFT ICLR 2023 | $\mathbf{31.0}_{\pm 1.5}$ | $36.5_{\pm 3.1}$ | $34.5_{\pm 1.1}$ | 4.01 | 3.98 | 3.72 | 3.35 |
| CLAMP ICLR 2023 | $26.1_{\pm 1.3}$ | $37.0_{\pm 2.0}$ | $34.2_{\pm 1.2}$ | 3.42 | 3.19 | 3.28 | 3.42 |
| Linear Probe | $29.0_{\pm 1.4}$ | $38.9_{\pm 1.8}$ | $36.8_{\pm 1.1}$ | 2.98 | 3.55 | 4.18 | 4.81 |
| Quadratic Probe | $28.8_{\pm 1.4}$ | $\mathbf{39.2}_{\pm 1.8}$ | $\mathbf{37.1}_{\pm 1.0}$ | **2.96** | **3.02** | 3.31 | 3.46 |

our method or using gradient descent steps to minimize the cross-entropy loss (Free-opt). Our quadratic probe (q-probe) outperforms the Free-Optimisation approach, and is less sensitive to the number of epochs the model should be optimized with. This sensitivity to the number of epochs becomes especially problematic under domain shifts, as this hyperparameter is fitted on a validation set that is not representative of the test set. Finally, by adding a regularization on the norm of $\mathbf{M}_k$ to the Free-Opt approach, the model becomes less sensitive to the number of epochs but achieves lower performances than our proposed optimization approach.

### 4.3 Impact of Domain Shifts

To further investigate the robustness of the different methods, we wish to evaluate the models' performances when the tasks they are evaluated on differ from the distribution of the training set. We designed scenarios extracted from the therapeutic data commons platform (Huang et al., 2021) and the LIT-PCBA dataset (Tran-Nguyen et al., 2020), in which the tasks yield a domain shift compared to the base dataset.

First, the FS-mol dataset mainly comprises balanced tasks, as explained in the previous section, while the tasks we will consider in this assessment are more imbalanced. The robustness of those different methods to class imbalance is essential, as molecules are often inactive in practice.

In this section, we focus our work on the top-performing models on FS-mol, the linear and quadratic probes (**l-probe**, **q-probe**), **protonet**, ADKF-IFT **(adkt)**, **clamp**, and the standard chemo-informatic method: **simsearch**.

### 4.3.1 QSAR modelling with imbalanced class distribution

**Datasets.** In this section, we will consider the Drug-Target-Interaction datasets available on the Therapeutic Data Common (Huang et al., 2021) platform. Drug Target interaction refers to a family of tasks that aim at predicting the affinity of a molecule to a given target using the molecular graph of the molecule, and the sequence of the target (which we discarded in our experiments)

The datasets chosen to evaluate the models are the KIBA (Tang et al., 2014), DAVIS (Davis et al., 2011), and BindingDB_Kd (Liu et al., 2006) datasets. These datasets are mainly composed of kinases, the most common target type in the FS-mol benchmark. Hence, this section will help evaluate the robustness of the different methods to a prior shift (difference in the label's marginal distribution), keeping the target of the task in-domain. The mean class imbalance, measured as the absolute deviation of the proportion of positive examples to a perfectly balanced scenario, is the lowest in the KIBA dataset (0.12), while the BindingDB_Kd and DAVIS datasets possess more imbalanced tasks (0.19 and 0.21, respectively).

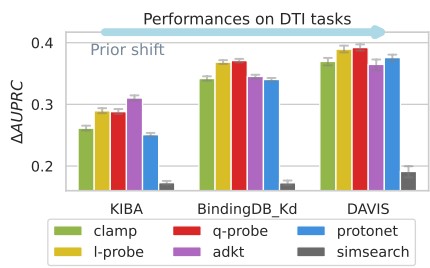

Figure 3: Performances of various methods on the DTI tasks. The fine-tuning baselines outperform the meta-learning methods when the tasks become more imbalanced.

**Evaluation procedure.** As some tasks are composed of a few molecules (all tasks in the DAVIS dataset are composed of 68 molecules), we only considered support sets of 16 molecules in this evaluation. All models were trained on the FS-mol benchmark, and their hyperparameters were also chosen on this benchmark. We used the $\Delta AUCPR$ metric to evaluate our models.

**Results.** The results of the different methods on those datasets are presented in Figure 3. While the state-of-the-art meta-learning methods achieve competitive results on the KIBA dataset, our baselines outperform all other methods when the distribution of the labels becomes more imbalanced. The performances of the linear and quadratic probes are similar, which is expected as we performed this evaluation on a support set consisting of 16 molecules, a setup where both methods achieved similar results on the main benchmark. These results seem to indicate that the fine-tuning baselines are more robust to prior shifts.

On the other hand, experiments on the BindingDB_Ki and IC50 datasets are presented in the Appendix A.5.1, where our fine-tuning baselines do not outperform meta-learning models, but where no model significantly outperforms the similarity search baseline.

### 4.3.2 Library screening

**Datasets.** Finding potential hits (active molecules) in a large library of compounds, referred to as library screening, is a common step in drug discovery, where scientists can then buy the most promising compounds from a commercial molecular database (Gentile et al., 2022). In these tasks, the proportion of "hits" (active compounds) is usually very low ($\approx 1\%$), yielding a much more significant prior shift than in the previous section. Besides, on the opposite of the section above, the targets considered in the datasets we will use are not kinases (see Appendix A.5.2), including another type of domain shift. To perform this evaluation, we will use 6 HTS datasets from the therapeutic data commons platform (Huang et al., 2021) and ten datasets from the LIT-PCBA database (Tran-Nguyen et al., 2020); details on the task choice can be found in the Appendix A.5.2.

**Evaluation procedure.** The experiments were run by considering support sets of 16, 32, 64 and 128 molecules, with a proportion of hits of both 5% and 10% in the support set, leaving all remaining hits in the query set. We considered the top-k% hitrate (proportion of true positive in the predicted positive when selecting k-% of the library) as a metric. Taking the average hitrate over all datasets can be problematic, as the average performance of the models largely varies from one dataset to the other. We hence decided to report the average rank of each model using autorank (Herbold, 2020), over 20 runs.

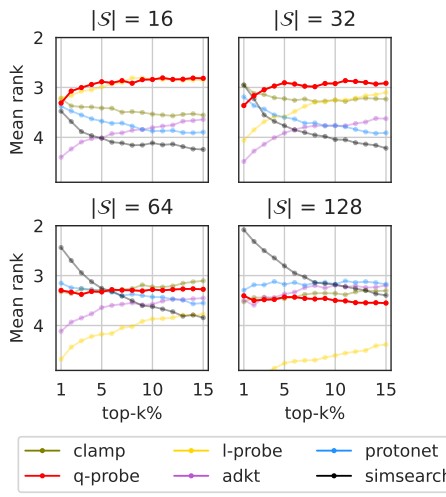

Figure 4: Average ranking performances of each method on the HTS tasks according to the percentage of the dataset selected. The quadratic probe obtains the best results when the support set's size is small, while the similarity search is the best model with larger support sets.

**Results.** Figure 4 presents the average rank of each model on the HTS tasks over 20 runs, for the different support set sizes, and proportion of molecules selected.

First, with a limited number of known labels (support set size of 16 or 32), our method consistently outperforms all other baselines, a trend observed across varying proportions of selected molecules. CLAMP also proves to be competitive in this low-data regime, surpassing the performance of meta-learning methods and the linear probe with larger support sets. Indeed, the linear probe exhibits strong performance with a support set size of 16, but its performance significantly diminishes as the support set size increases.

As the support set size increases, meta-learning methods demonstrate comparable results to the quadratic probe and CLAMP, until protonet outperforms the quadratic probe with a support set size of 128. Notably,

the similarity search, employed as our baseline for these experiments, is outperformed by most other methods when the support set comprises only a few molecules. However, it becomes the best model when the support set is extensive, exhibiting superior performances, especially when few molecules are selected in each dataset (1 to 5%).

The robustness of the similarity search in these tasks can be attributed to the fact that it does not rely on any pre-training, and is hence not affected by the domain shift between the HTS and FS-mol datasets.

## 5 Summary and Concluding Remarks

While meta-learning methods consistently yield highly competitive results in molecular few-shot learning, it is essential to acknowledge that fine-tuning baselines have the potential to achieve comparable outcomes. In particular, we explored the intricacies of optimizing an extension of the linear probe, a quadratic probe. Both probing methods are capable of competing with state-of-the-art meta-learning methods while relying on a model pre-trained with a standard multitask objective.

Moreover, our findings demonstrate that these fine-tuning baselines exhibit heightened robustness when confronted with a shift in the label distribution. This resilience further enhances their practical utility in diverse scenarios. Although the quadratic probe requires an additional step—estimating covariance matrices—it consistently yields slightly better results than the linear probe, with notable improvements under certain domain shifts. Thus, both methods provide efficient few-shot classifiers with distinct strengths that can be evaluated based on the demands of specific real-world applications.

Regrettably, these baseline approaches are often overlooked in the existing literature in this field. Notably, these methods possess distinctive advantages, particularly in a black-box setting, where they can be readily benchmarked upon the availability of a pre-trained model.

The anonymized codebase used in this paper is available at `https://github.com/Fransou/Strong-Baseline-Molecular-FSL`

## 6 Statement of Broader Impact

This paper presents work that aims to advance the field of Machine Learning. Our work has many potential societal consequences. We hope our work can be used to help drug discovery pipelines, help discover treatments, and accelerate the process. We did not identify any risk of harm specific to our work.

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

# A Appendix

## A.1 Algorithm

---

**Procedure 1** Training of the quadratic probe

---

**Input:** A support set $\mathcal{S} = \{\mathbf{z}_i, y_i\}_{i \leq N}$, the model's weights $\{\mathbf{w}_k\}_{k \in \{0,1\}}$, a learning rate $\alpha$ and a shrinkage coefficient $\lambda$.

  **for** $i \in \{1, \ldots, N_{epochs}\}$ **do**
    **for** $k \in \{0, 1\}$ **do**
      $\mathcal{S}_k \leftarrow \{\mathbf{z}_i | y_i = k\}_{i \leq N}$
      $\mathbf{M}_k \leftarrow \left( (1 - \lambda) \frac{1}{|\mathcal{S}_k|} \sum_{\mathbf{z} \in \mathcal{S}_k} (\mathbf{z} - \mathbf{w}_k)(\mathbf{z} - \mathbf{w}_k)^T + \lambda \mathbf{I} \right)^{-1}$ {Compute the covariance matrices}
    **end for**
    $\Theta \leftarrow \{\mathbf{w}_k, \mathbf{M}_k\}_{k \in \{0,1\}}$
    $\mathcal{L}_{ce}(\Theta) \leftarrow \frac{1}{|\mathcal{S}|} \sum_{\mathbf{z}, y \in \mathcal{S}} \mathcal{L}_{ce}(\mathbf{z}, y, \Theta)$
    **for** $k \in \{0, 1\}$ **do**
      $\mathbf{w}_k \leftarrow \mathbf{w}_k - \alpha \nabla_{w_k} \mathcal{L}_{ce}(\Theta)$ {Update the weights with gradient descent}
    **end for**
  **end for**
  Return $\Theta$

---

## A.2 Proofs

**Proof of Proposition** 3.1

*Proof.* The samples from both classes are linearly separable. As such, there exists a vector $\boldsymbol{v}$ such that

$$\forall i \in \{0, \ldots, N\}, \begin{cases} \boldsymbol{v}^T \boldsymbol{z}_i > b & \text{if } y_i = 1 \\ \boldsymbol{v}^T \boldsymbol{z}_i < b & \text{if } y_i = 0, \end{cases} \tag{8}$$

where $b$ is a constant, and we take $\boldsymbol{v}$ such that $\|\boldsymbol{v}\| = 1$.

First, by dividing the expression of $p_\Theta(Y = y | Z = \boldsymbol{z})$ by its numerator:

$$\mathcal{L}_{ce}(\boldsymbol{z}_i, y_i, \Theta) = \log(1 + e^{-(\|\boldsymbol{z}_i - \boldsymbol{w}_{1-y_i}\|^2_{\mathbf{M}_{1-y_i}} - \|\boldsymbol{z}_i - \boldsymbol{w}_{y_i}\|^2_{\mathbf{M}_{y_i}})}) \geq 0. \tag{9}$$

**Our goal, is to build a set of parameters $\Theta(\lambda)$ such that $\lim_{\lambda \to +\infty} \mathcal{L}_{ce}(\boldsymbol{z}_i, y_i, \Theta(\lambda)) = 0$, and $\lambda$ is an eigenvalue of $\mathbf{M}_k$ for all $k \in \{0, 1\}$.**

Since $\Sigma_k$ is PSD, we can decompose it as $\mathbf{M}_k = \mathbf{U}_k^t \Lambda_k \mathbf{U}_k$, with $\mathbf{U}_k$ an orthogonal matrix, and $\Lambda_k = \text{diag}(\lambda_{k,0}, \ldots \lambda_{k,d})$ a diagonal matrix containing the eigenvalues of $\mathbf{M}_k$. The matrix $\mathbf{U}_k$ corresponds to a rotation of the vector space to a new basis corresponding to the eigenvectors of $\mathbf{M}_k$ (the rows of $\mathbf{U}_k$).

We choose $\mathbf{M}_k(\lambda)$ so that its first eigenvector is $\boldsymbol{v}$ with an eigenvalue of $\lambda_{k,0} = \lambda$, and we keep all other eigenvalues and eigenvector constant. We set $\boldsymbol{w}_k$ as:

$$\boldsymbol{w}_k = \begin{cases} (b - 1)\boldsymbol{v} & \text{if } k = 0 \\ (b + 1)\boldsymbol{v} & \text{if } k = 1. \end{cases}$$

First, let's find an equivalent of the Mahalanobis distance when $\lambda \to +\infty$:

$$\|\boldsymbol{z}_i - \boldsymbol{w}_k\|^2_{\mathbf{M}_k} = \frac{1}{2}(\boldsymbol{z}_i - \boldsymbol{w}_k)^T \mathbf{U}^T \Lambda_k \mathbf{U} (\boldsymbol{z}_i - \boldsymbol{w}_k)$$

$$\sim_{\lambda \to +\infty} \frac{\lambda}{2}(\boldsymbol{z}_i - \boldsymbol{w}_k)^T \boldsymbol{v}\boldsymbol{v}^T (\boldsymbol{z}_i - \boldsymbol{w}_k)$$

$$\sim_{\lambda \to +\infty} \frac{\lambda}{2}(\boldsymbol{v}^T \boldsymbol{z}_i - \boldsymbol{v}^T \boldsymbol{w}_k)^2 .$$

Let's start with the case $y_i = 1$:

$$\|\boldsymbol{z}_i - \boldsymbol{w}_{1-y_i}\|^2_{\mathbf{M}_{1-y_i}} \sim_{\lambda \to +\infty} \frac{\lambda}{2}\left(1 + \boldsymbol{v}^T\boldsymbol{z}_i - b\right)^2$$

$$\|\boldsymbol{z}_i - \boldsymbol{w}_{y_i}\|^2_{\mathbf{M}_{y_i}} \sim_{\lambda \to +\infty} \frac{\lambda}{2}\left(1 - \left(\boldsymbol{v}^T\boldsymbol{z}_i - b\right)\right)^2.$$

Since we chose $y_i = 1$, the assumption of linear separability in the equation 8 states that $\boldsymbol{v}^T\boldsymbol{z}_i - b > 0$. As a result, $\left(1 + \boldsymbol{v}^T\boldsymbol{z}_i - b\right)^2 > \left(1 - \left(\boldsymbol{v}^T\boldsymbol{z}_i - b\right)\right)^2$. Hence, for all $i$ such that $y_i = 1$,

$$\left(\boldsymbol{v}^T\left(\boldsymbol{z}_i - \boldsymbol{w}_{1-y_i}\right)\right)^2 > \left(\boldsymbol{v}^T\left(\boldsymbol{z}_i - \boldsymbol{w}_{y_i}\right)\right)^2. \tag{10}$$

The same reasoning applies for $y_i = 0$, and we finally obtain for all $i \in \{0, \dots, N\}$:

$$\lim_{\lambda \to +\infty} \|\boldsymbol{z}_i - \boldsymbol{w}_{1-y_i}\|^2_{\mathbf{M}_{1-y_i}} - \|\boldsymbol{z}_i - \boldsymbol{w}_{y_i}\|^2_{\mathbf{M}_{y_i}} = \lim_{\lambda \to +\infty} \frac{\lambda}{2}\left(\left(\boldsymbol{v}^T\left(\boldsymbol{z}_i - \boldsymbol{w}_{1-y_i}\right)\right)^2 - \left(\boldsymbol{v}^T\left(\boldsymbol{z}_i - \boldsymbol{w}_{y_i}\right)\right)^2\right)$$
$$= +\infty. \tag{11}$$

Finally, using the equation 11, and the equation 9, we obtain:

$$\lim_{\lambda \to +\infty} \mathcal{L}_{ce}(\boldsymbol{z}_i, y_i, \Theta(\lambda)) = 0. \tag{12}$$

Note that this proof holds because we could choose a direction (an eigenvector of $\mathbf{M}_k$) common to all data points, obtained from the linear separability assumption on the data. □

**Proof of Remark** 3.2

*Proof.* It is straightforward to see that for a given data point $\boldsymbol{z}_i$, $\mathcal{L}_{ce}(\boldsymbol{z}_i, y_i, \Theta) = f_1\left(\boldsymbol{z}_i, y_i, \Theta\right) + f_2\left(\boldsymbol{z}_i, \Theta\right)$.

Let's first consider $f_1$. Since $\mathbf{M}_k$ is PSD,

$$0 \le \|z_i - \boldsymbol{w}_k\|^2_{\mathbf{M}_k} = \|D_k\left(z_i - \boldsymbol{w}_k\right)\|^2_2 \le \|D_k\|^2_F\|z_i - \boldsymbol{w}_k\|^2_2. \tag{13}$$

Hence when $\|\mathbf{M}_k\|_F \to 0$, $f_1 \to 0$.

Then, let's consider $f_2$.

$$\underset{\mathbf{M}_k, k \in \{0,1\}}{\arg\min} f_2\left(\boldsymbol{z}_i, \Theta\right) = \underset{\mathbf{M}_k, k \in \{0,1\}}{\arg\min} \log\left(\sum_{k \in \{0,1\}} \exp\left(-\|\boldsymbol{z}_i - \boldsymbol{w}_k\|^2_{\mathbf{M}_k}\right)\right)$$
$$= \underset{\mathbf{M}_k, k \in \{0,1\}}{\arg\min} \sum_{k \in \{0,1\}} \exp\left(-\|\boldsymbol{z}_i - \mathrm{w}_k\|^2_{\mathbf{M}_k}\right)$$
$$= \left\{\underset{\mathbf{M}_k}{\arg\min} \exp\left(-\|\boldsymbol{z}_i - \mathrm{w}_k\|^2_{\mathbf{M}_k}\right)\right\}_{k \in \{0,1\}}.$$

Minimizing $f_2$ is then equivalent to maximizing $\|\boldsymbol{z}_i - \mathrm{w}_k\|^2_{\mathbf{M}_k}$ for all $k \in \{0, 1\}$, and:

$$\|z_i - \boldsymbol{w}_k\|^2_{\mathbf{M}_k} = (\boldsymbol{z}_i - \boldsymbol{w}_k)^T\mathbf{M}_k(\boldsymbol{z}_i - \boldsymbol{w}_k) \ge \sigma_{k,min}\|\boldsymbol{z}_i - \boldsymbol{w}_k\|^2_2, \tag{14}$$

where $\sigma_{k,min}$ is the minimal eigenvalue of $\mathbf{M}_k$.

As a result, $f_2$ is minimized when for all $k \in \{0, 1\}$ $\sigma_{k,min} \to +\infty$, and hence when $\|\mathbf{M}_k\|^2_F \to +\infty$.

This decomposition illustrates the role of each term in the cross-entropy loss, the first trying to reduce the distance between each point belonging to the same class, the other increasing the distance between all points. □

**Proof of Proposition** 3.3

*Proof.* For a molecular embedding $\boldsymbol{z}_i$, the expression of $f_2(\boldsymbol{z}_i, \Theta)$ is:

$$f_2(\boldsymbol{z}_i, \Theta) = \log\left(\sum_{k \in \{0,1\}} \exp\left(-\|\boldsymbol{z}_i - \boldsymbol{w}_k\|^2_{\mathbf{M}_k}\right)\right).$$

We use the following property of the log-sum-exp function:

$$\max\left\{-\|\boldsymbol{z}_i - \boldsymbol{w}_k\|^2_{\mathbf{M}_k}\right\}_{k \in \{0,1\}} \leq \log\left(\sum_{k \in \{0,1\}} \exp\left(-\|\boldsymbol{z}_i - \boldsymbol{w}_k\|^2_{\mathbf{M}_k}\right)\right) \leq \max\left\{-\|\boldsymbol{z}_i - \boldsymbol{w}_k\|^2_{\mathbf{M}_k}\right\}_{k \in \{0,1\}} + \log 2.$$

We can then majorize the maximum function:

$$|\max\left\{-\|\boldsymbol{z}_i - \boldsymbol{w}_k\|^2_{\mathbf{M}_k}\right\}_{k \in \{0,1\}}| = \min\left\{\|\boldsymbol{z}_i - \boldsymbol{w}_k\|^2_{\mathbf{M}_k}\right\}_{k \in \{0,1\}} \leq \frac{1}{2}\sum_{k \in \{0,1\}}\|\boldsymbol{z}_i - \boldsymbol{w}_k\|^2_{\mathbf{M}_k}.$$

Furthermore, $\|\boldsymbol{z}_i - \boldsymbol{w}_k\|^2_{\mathbf{M}_k} = \|D_k(\boldsymbol{z}_i - \boldsymbol{w}_k)\|^2_2$.

And $\|D_k(\boldsymbol{z}_i - \boldsymbol{w}_k)\|^2_2 \leq \|D_k\|^2_F\|\boldsymbol{z}_i - \boldsymbol{w}_k\|^2_2$, where $\|\boldsymbol{z}_i - \boldsymbol{w}_k\|^2_2$ is constant w.r.t $\mathbf{M}_k$,

$$f_2(\boldsymbol{z}_i, \Theta) = O\left(\sum_{k \in \{0,1\}}\|D_k\|^2_F\right) = O\left(\sum_{k \in \{0,1\}}\|\mathbf{M}_k\|_F\right).$$

Let's study the asymptotic behavior of $\tilde{f}_2(\boldsymbol{z}_i, \Theta)$.

$$|\tilde{f}_2(\boldsymbol{z}_i, \Theta)| = |\sum_{k \in \{0,1\}}\frac{|\mathcal{S}_k|}{|\mathcal{S}|}\log\det\mathbf{M}_k|$$

$$= |\sum_{k \in \{0,1\}}\frac{|\mathcal{S}_k|}{|\mathcal{S}|}\sum_{j \in \{1...d\}}\log\sigma_{k,j}|$$

$$\leq d\sum_{k \in \{0,1\}}\frac{|\mathcal{S}_k|}{|\mathcal{S}|}\max\{|\log\sigma_{k,j}|\}_{j \in \{1...d\}}.$$

Where d is the dimension of the data, and $\sigma_{k,j}$ is the $j$-th eigenvalue of $\mathbf{M}_k$.

Since all eigenvalues of $\mathbf{M}_k$ for $k \in \{0,1\}$ grow to infinity, $\max\{|\log\sigma_{k,j}|\}_{j \in \{1...d\}} = O(\log\max\{\sigma_{k,j}\}_{j \in \{1...d\}})$.

Finally, all precision matrices are PSD, hence for $k \in \{0,1\}$: $\sqrt{\sum_{j \in \{1...d\}}\sigma^2_{k,j}} = \|\mathbf{M}_k\|_F \geq \max\{\sigma_{k,j}\}_{j \in \{1...d\}}$.

As a result $\tilde{f}_2(\boldsymbol{z}_i, \Theta) = O\left(\sum_{k \in \{0,1\}}\log\|\mathbf{M}_k\|_F\right)$.

$\square$

**Proof of Remark** 3.4

*Proof.* First the log-likelihood of a gaussian distribution of mean $\boldsymbol{w}_k$ and of precision matrix $\mathbf{M}_k$, evaluated on $\boldsymbol{z}$ is:

$$\mathcal{L}_{\boldsymbol{w}_k, \mathbf{M}_k}(\boldsymbol{z}) = -\frac{1}{2}\left(\log\det\mathbf{M}_k - \|\boldsymbol{z} - \boldsymbol{w}_k\|^2_{\mathbf{M}_k}\right) + c,$$

where c is a constant.

For a given support set $\mathcal{S}$, let's compute the modified loss:

$$\frac{1}{|\mathcal{S}|}\sum_{i\in\mathcal{S}}\left(f_1(\boldsymbol{z}_i,y_i,\Theta)+\tilde{f}_2(\Theta)\right) = \frac{1}{|\mathcal{S}|}\sum_{i\in\mathcal{S}}\left(\|\boldsymbol{z}-\boldsymbol{w}_{y_i}\|^2_{\mathbf{M}_{y_i}}-\sum_{k\in\{0,1\}}\frac{|\mathcal{S}_k|\log\det\left(\mathbf{M}_k\right)}{|\mathcal{S}|}\right)$$

$$= \frac{1}{|\mathcal{S}|}\sum_{i\in\mathcal{S}}\|\boldsymbol{z}-\boldsymbol{w}_{y_i}\|^2_{\mathbf{M}_{y_i}}-\frac{1}{|\mathcal{S}|}\sum_{k\in\{0,1\}}|\mathcal{S}_k|\log\det\left(\mathbf{M}_k\right)$$

$$= \frac{1}{|\mathcal{S}|}\sum_{k\in\{0,1\}}\sum_{i\in\mathcal{S}_k}\left(\|\boldsymbol{z}-\boldsymbol{w}_k\|^2_{\mathbf{M}_k}-\log\det\left(\mathbf{M}_k\right)\right).$$

For a class $k \in \{0,1\}$, we are trying to minimize the negative log-likelihood of a Gaussian distribution evaluated on the support set $\mathcal{S}_k$, with mean $\boldsymbol{w}_k$ and precision matrix $\mathbf{M}_k$.

$\square$

### A.3 Divergence of $\mathbf{M}_k$

We study the evolution of the value of $\mathbf{M}_k$ when optimized with gradient descent (Free-Opt) and using the quadratic probe (q-probe). The results are reported in Figure 5, where we see that the maximal eigenvalue of $\mathbf{M}_k$, which diverges with gradient descent, while remaining stable using the quadratic probe (we recall that the molecule's embedding are distributed on the unit sphere of $\mathbb{R}^d$).

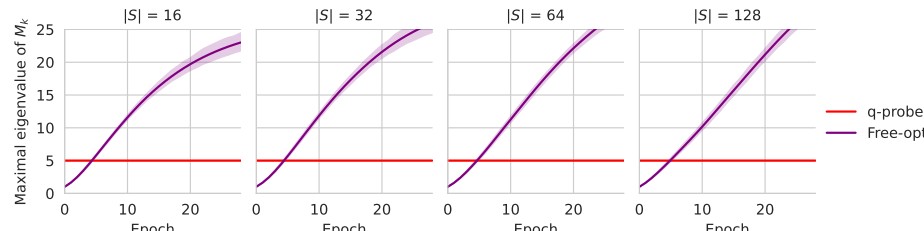

Figure 5: Evolution of the maximal eigenvalue of $\mathbf{M}_k$ along training when optimized with gradient descent (Free-Opt) or with the quadratic probe (q-probe).

### A.4 Effect of $\lambda$ on the quadratic probe's performances

Estimating the covariance matrix of the data is challenging in the few-shot setting. To regularize the empirical covariance matrix, we use a diagonal shift to make the inverse of the covariance matrix well-defined and more stable. The optimal value of $\lambda$ varies with the size of the support set, as shown in the Figure 6. We see that the optimal value of $\lambda$ decreases with the size of the support set, which is expected as the empirical covariance matrix becomes more stable with more samples.

All hyperparameters of the meta-learning methods are constant over the support set sizes. In particular, all meta-learning methods are trained with a fixed support set size (64 for Protonet, for instance) and evaluated on all support set sizes. We chose similarly to use a single value of $\lambda = 0.2$ across all support set sizes. However, while modifying a hyperparameter in a meta-learning method usually requires retraining the model, only the few-shot adaptation step needs to be re-run when changing $\lambda$ in the quadratic probe, for instance.

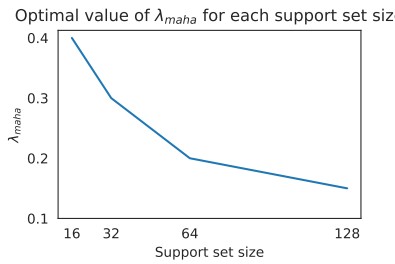

Figure 6: Optimal values of $\lambda_{maha}$ found during the hyperparameter search on FS-mol's validation set. (computing $\Sigma_k \triangleq (1-\lambda)\Sigma_{k,\text{empirical}} + \lambda\mathbf{I}$)

## A.5    Out-of-domain experiments details

Some few-shot learning methods for drug discovery proposed out-of-
domain scenarios to evaluate their models. Still, the domain shifts involved are often unrealistic, such as evaluating a model on Tox21 (Huang et al., 2016), a dataset containing chemicals that are not drug-like (pesticides, food additives, . . . ), or evaluating models' performances on material design tasks. However, models trained and designed for activity prediction would marginally be used in such drastically different scenarios, giving limited insights into the model's performances. To propose more realistic domain-shift experiments, we decided to consider scenarios using molecular inputs that do not drastically differ from the ones seen in training, as well as considering tasks related to activity prediction.

### A.5.1    Drug-target-interaction details

**Creation of Few-shot tasks**

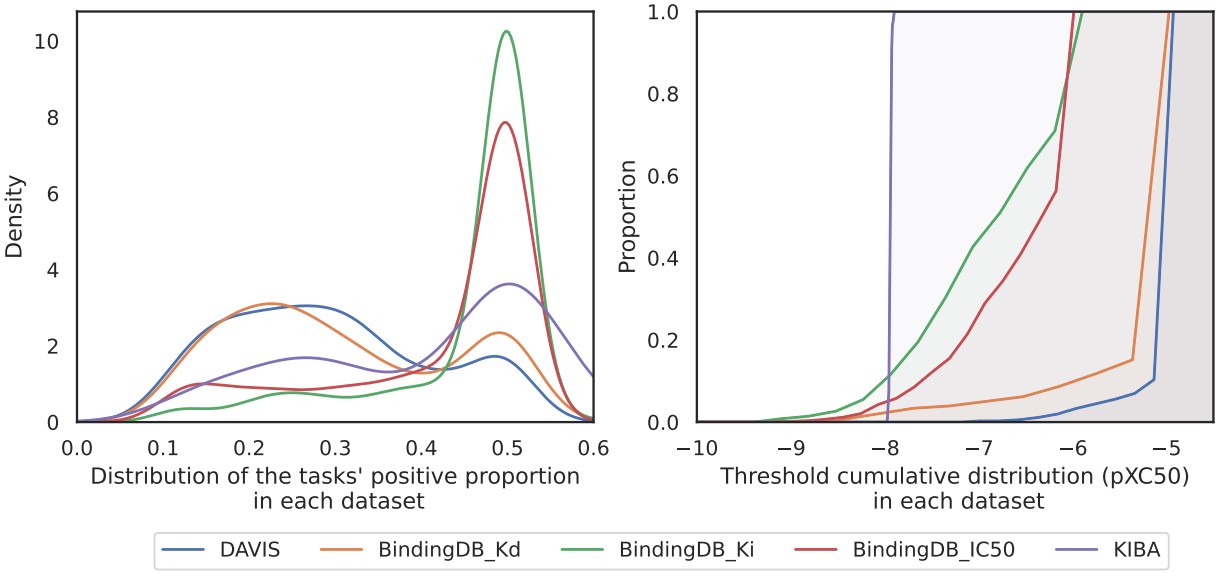

Figure 7: Imbalance of the DTI datasets' task classes and distribution of the thresholds used to obtain the classification labels in each dataset.

As explained in Section 4.3.1, we introduce various new datasets on which we compared the various methods. To adapt the various datasets to the few-shot learning scenario, we followed five steps:

1. Identifying tasks already in the FS-mol training set is not trivial, as the FS-mol dataset is large. To do so, we removed every task in the DTI datasets that contained more than 30% of molecules already in the FS-mol training set.

2. We then removed every SMILES that cannot be processed by the RDKit library or ended up with a molecule containing an atom with a valence greater than 6.

3. A molecule in a given task can be associated with several different activity values on the same target. We decided to remove any molecule associated with multiple measurements in a task.

4. We separated the molecules into active and inactive classes by computing the median of the activity values in the task, clipped between a maximum pXC50 of 9 and a minimum of pXC50 of 5, or 6 for BindingDB_Ki and BindingDB_IC50. We then removed molecules whose activity was equal to the threshold chosen in the previous step.

5. Finally, we dropped all tasks with less than 60 molecules, or more than 5000, and removed all tasks whose label distribution was too imbalanced (a positive proportion outside of the $[0.1, 0.6]$ range).

The resulting datasets provide different class imbalances, illustrated in the Figure 7, inferred by the choice of the thresholds on the activity numeric values for each task.

**Complementary results and limitations**

We provided in the paper the results on DAVIS, KIBA and BindingDB_Kd, and we provide supplementary results on BindingDB_IC50 and BindingDB_Ki in the Figure 8. We see that the fine-tuning methods we presented do not obtain competitive results, which implies that our pretraining procedure resulted in a model that is still sensible to shifts in the target type. Finally, CLAMP's fine-tuning outperforms our methods, which rely on a standard multi-task model. This can be explained by the fact that CLAMP is trained on textual descriptions of the assays. Obtaining textual information about the assays allows the model to assimilate knowledge that is shareable and applicable across various target types, potentially leading to better robustness to this domain shift.

ADKT-IFT obtains competitive results, showing strong robustness to the type of target considered, but none of the methods achieved significantly better results than the similarity search. This underscores the significant challenge for the presented few-shot methods in effectively adapting to new targets.

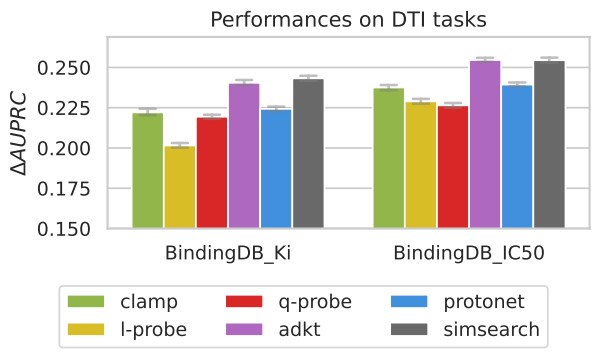

Figure 8: Performances of the considered methods on the BindingDB_Ki and BindingDB_IC50 datasets. Few-shot models do not significantly outperform the similarity search, and CLAMP is the best fine-tuning method.

### A.5.2 Library screening details

**Creation of Few-shot tasks**

This section provides more details on adapting the HTS datasets we used in our experiments. HTS datasets correspond to assays usually evaluated early in the drug discovery process. These tasks often comprise many compounds with very few active molecules to identify. We considered datasets available through the Therapeutic Data Commons (TDC) (Huang et al., 2021) plateform (Touret et al., 2020; Wu et al., 2018; Butkiewicz et al., 2013), and various tasks from the LIT-PCBA dataset (Tran-Nguyen et al., 2020), a database built on 149 dose–response PubChem bioassays (Wang et al., 2016).

Running these experiments on such a large number of molecules would be too computationally expensive. As a result, we down-sampled all datasets to 30000 molecules.

We processed each dataset as follows:

1. Only compounds whose SMILES can be processed by the RDKit library are kept, and we removed all molecules whose SMILES ended up with a molecule containing an atom with a valence greater than 6.

2. To create tasks of 30000 molecules, we first kept all active molecules if the resulting proportion of active molecules remained under 7%. Otherwise, we sampled the active molecules to keep the same proportion of active molecules as in the original dataset.

3. We then randomly sampled inactive molecules to obtain at most 30k molecules.

We removed all datasets initially composed of more than 300k compounds. We also did not include the dataset SARSCoV2_3CLPro_Diamond dataset, as it corresponds to a fragment screen, using a different

Table 3: Description of the datasets used for the library screening task.

| Benchmark | Dataset name name | Target type if known | Hit proportion | Dataset size |
|---|---|---|---|---|
| TDC | SARSCoV2 Vitro Touret | not known | 5.9% | 1500 |
| | HIV | not known | 4.8% | 30k |
| | Orexin1 Receptor | GPCR | 0.77% | 30k |
| | M1 Musarinic Receptor agonist | GPCR | 0.62% | 30k |
| | M1 Musarinic Receptor antagonist | GPCR | 1.2% | 30k |
| | CAV3 T-type Calcium channels | Ion Channel | 4.8% | 30k |
| LIT-PCBA | ALDH1 | oxidoreductase | 4.9% | 30k |
| | ESR1 antagonist | Nuclear receptor | 1.9% | 5k |
| | GBA | hydrolase | 0.54% | 30k |
| | MAPK1 | kinase | 1.0% | 30k |
| | MTORC1 | protein complex | 1.9% | 30k |
| | OPRK1 | GPCR | 0.066% | 30k |
| | PKM2 | kinase | 1.8% | 30k |
| | PPARG | nuclear receptor | 0.44% | 5k |
| | TP53 | antigen | 1.8% | 4k |
| | VDR | nuclear receptor | 2.9% | 30k |

input type than the drug-like compounds we considered so far. The Table 3 describes the datasets used for the library screening task. The datasets are diverse, with few datasets containing about 5% of hits (SARSCOV2_Vitro_Touret, HIV, CAV3_T-type_Calcium_channels, ALDH1), and the others having less than 2% of hits in general.

The Figure 9 reveals how different the molecular structure between hits can be in these datasets. In particular, we computed the Tanimoto similarity between all active compounds in each dataset. Also, we measure the maximum similarity between an active compound and the other hits in the task. In particular, the hits are structurally different in tasks such as PPARG or Orexin1 receptor. As a result, retrieving active molecules using the information of the hits available in the support set can be challenging. In some other tasks, such as OPRK1 or HIV, some hits share similar structures. The heterogeneity of the tasks motivates the choice of using rankings as a metric, as the models' performances on each task can be drastically different.

**Complete results**

Results on all datasets are presented in the figure 10 and 11, as a line plot displaying the evolution of the hitrate when the number of molecules selected increases. The datasets are ordered according to the value 75% quantile of the maximum Tanimoto similarity between an active compound and the other hits in the task. This can be interpreted as a mild approximation of the task's difficulty (as it does not consider similarity to inactive compounds or the proportion of hits).

Interestingly, in the estimated 'hardest' tasks, when the support set size increases, the performances of some models (such as the linear probe) do not always increase (such as on the SARSCOV2_Vitro_Touret or the ALDH1 datasets). This can be explained by the fact that the hits are structurally different. Indeed, adding a new hit to the support set can be seen as adding noise to the support set, adding a molecule whose structure does not help to find a common representation for all hits. This is especially the case on the SARSCOV2_Vitro_Touret dataset.

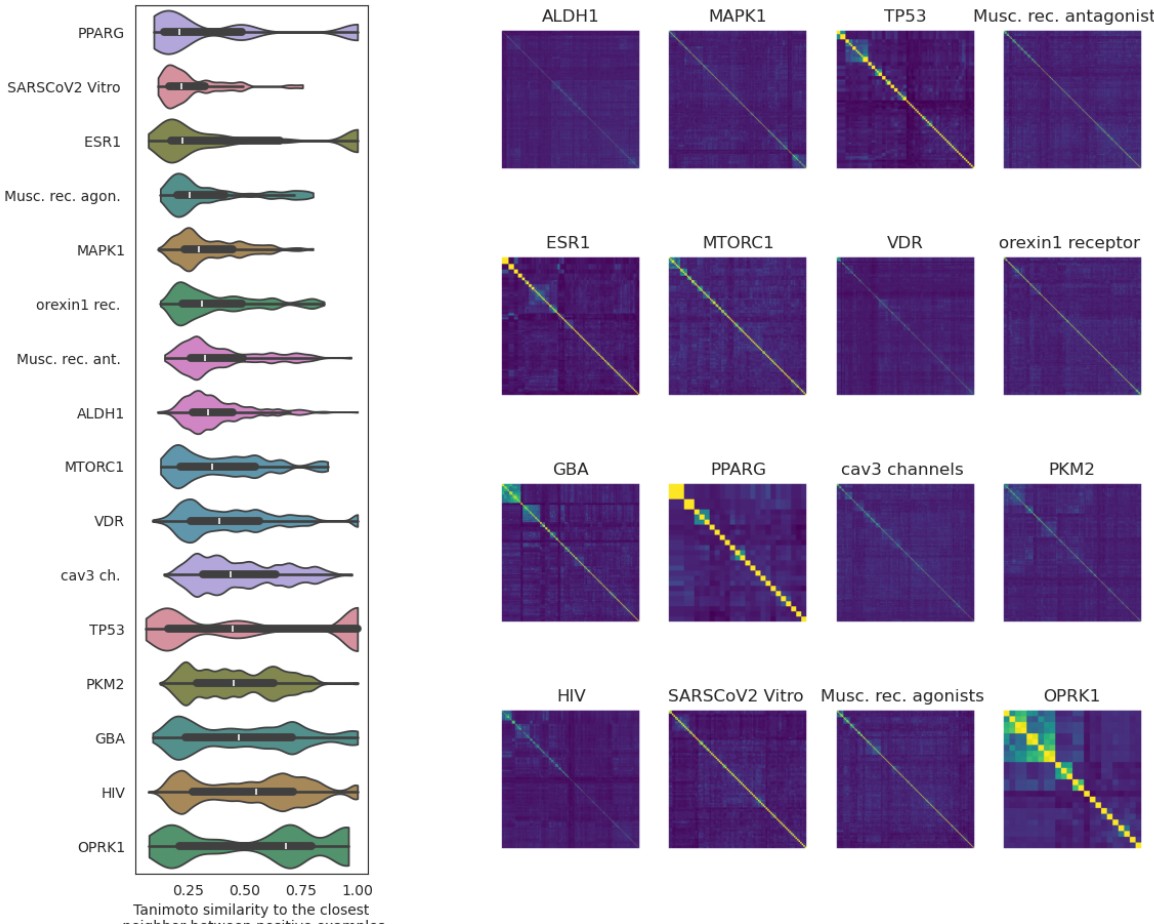

Figure 9: Distribution of the Tanimoto similarity between an active compound and its nearest active neighbour in the HTS datasets (left). Similarity matrices between the active compounds of the HTS datasets (right).

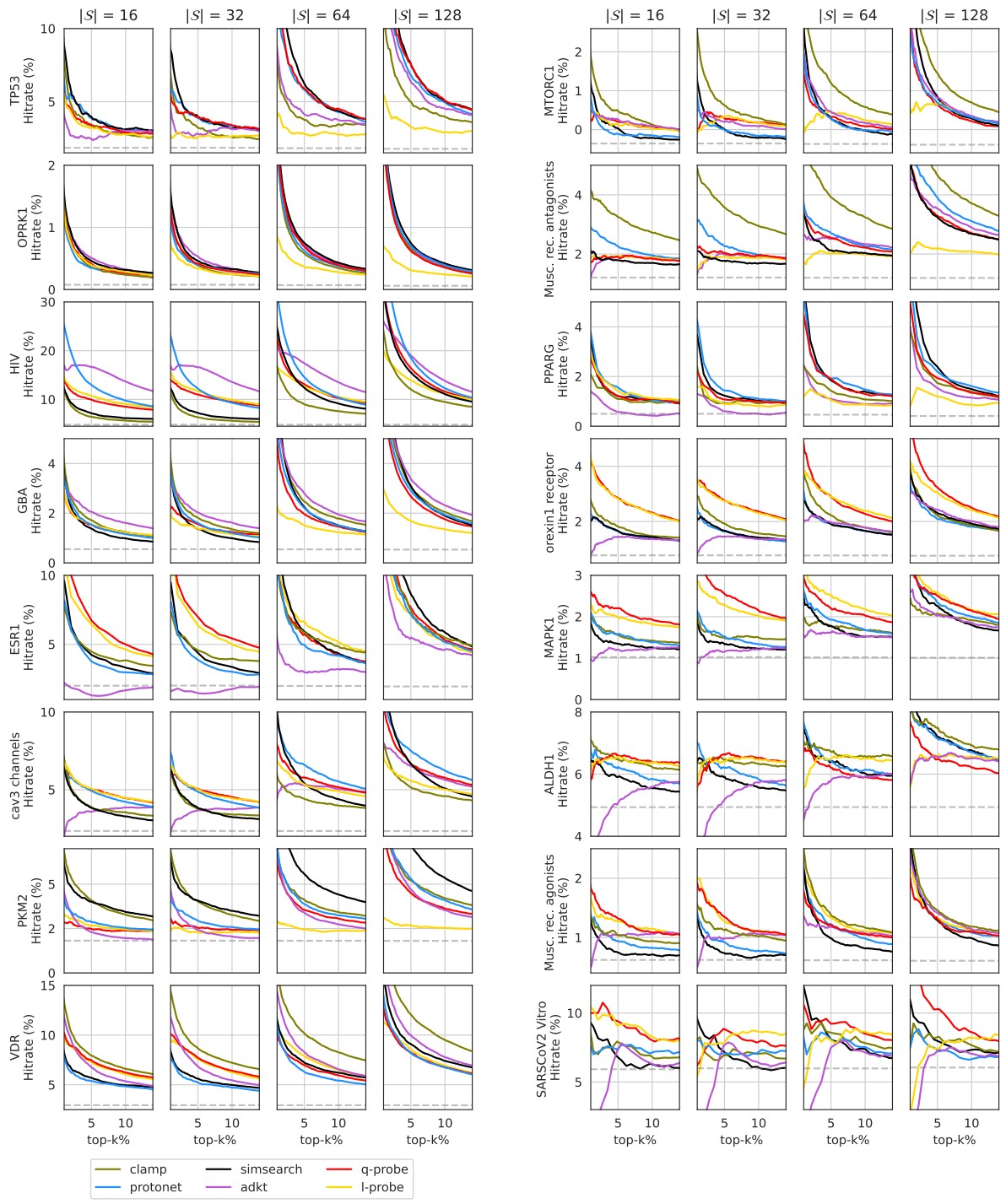

Figure 10: Evolution of the hitrate with the percentage of the dataset selected, on each dataset for different support set sizes, when 5% of the support set are hits.

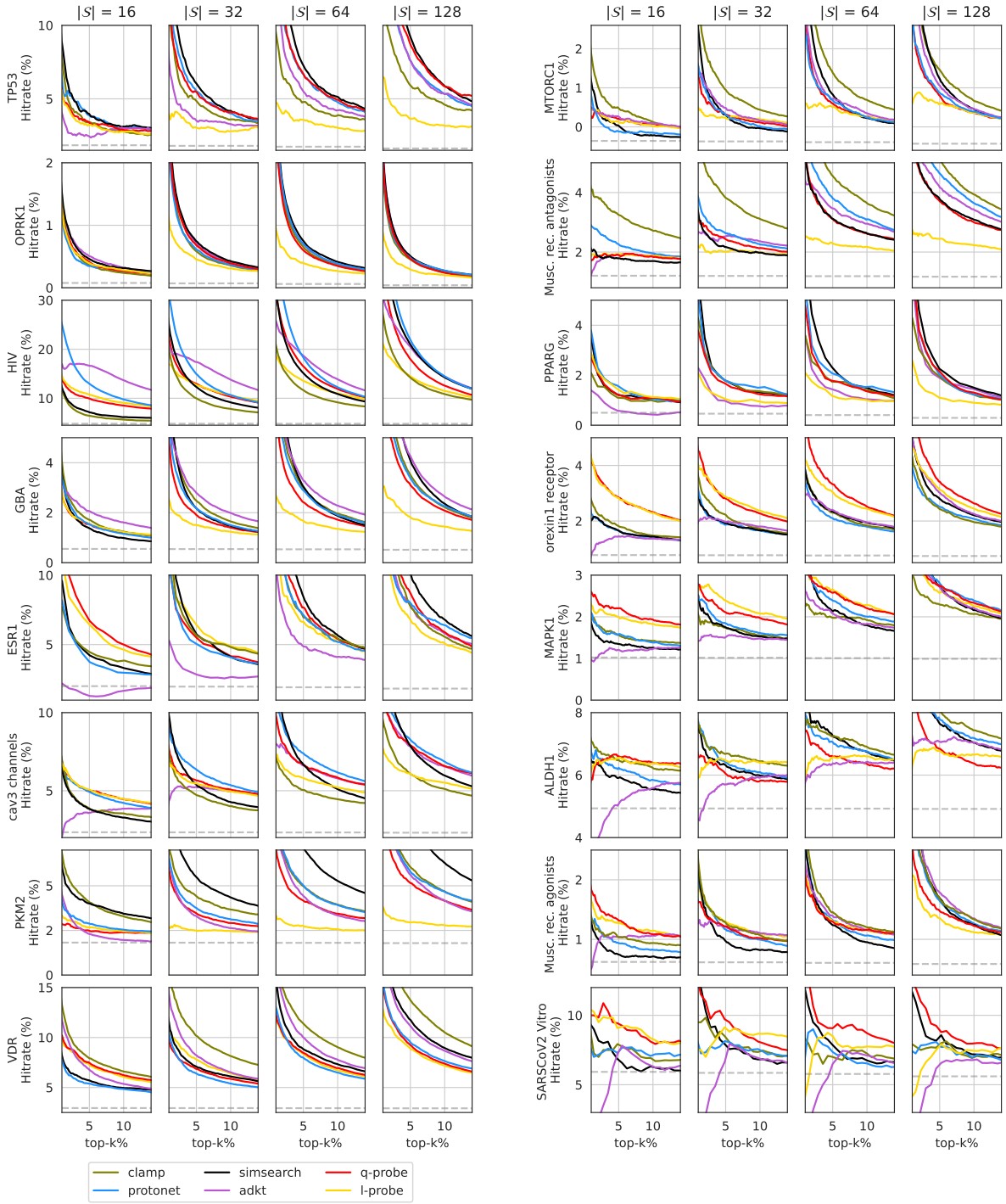

Figure 11: Evolution of the hitrate with the percentage of the dataset selected, on each dataset for different support set sizes, when 10% of the support set are hits.

## A.6 Performances of other molecular pretrained backbones

Table 4: Performance comparison of pretrained with different support set sizes trained with a linear probe, compared to the linear probe applied to our multi-task backbone.

| Model | $|\mathcal{S}| = 16$ | $|\mathcal{S}| = 32$ | $|\mathcal{S}| = 64$ | $|\mathcal{S}| = 128$ |
|---|---|---|---|---|
| MolBert (Fabian et al., 2020) | $0.045_{\pm 0.005}$ | $0.058_{\pm 0.005}$ | $0.074_{\pm 0.006}$ | $0.105_{\pm 0.007}$ |
| AttributeMask (Sun et al., 2022) | $0.048_{\pm 0.005}$ | $0.060_{\pm 0.005}$ | $0.077_{\pm 0.006}$ | $0.110_{\pm 0.007}$ |
| ChemBerta-MLM (Ahmad et al., 2022) | $0.049_{\pm 0.005}$ | $0.063_{\pm 0.006}$ | $0.080_{\pm 0.006}$ | $0.113_{\pm 0.007}$ |
| ThreeDInfomax (Stärk et al., 2021) | $0.050_{\pm 0.004}$ | $0.067_{\pm 0.005}$ | $0.081_{\pm 0.006}$ | $0.108_{\pm 0.006}$ |
| ContextPred (Sun et al., 2022) | $0.052_{\pm 0.005}$ | $0.067_{\pm 0.005}$ | $0.082_{\pm 0.006}$ | $0.113_{\pm 0.007}$ |
| GPT-GNN (Hu et al., 2020) | $0.056_{\pm 0.005}$ | $0.070_{\pm 0.005}$ | $0.085_{\pm 0.006}$ | $0.114_{\pm 0.006}$ |
| MolFormer (Ross et al., 2022) | $0.058_{\pm 0.006}$ | $0.075_{\pm 0.006}$ | $0.092_{\pm 0.007}$ | $0.125_{\pm 0.008}$ |
| MAT (Maziarka et al., 2019) | $0.052_{\pm 0.005}$ | $0.069_{\pm 0.006}$ | $0.092_{\pm 0.007}$ | $0.136_{\pm 0.009}$ |
| GROVER (Rong et al., 2020) | $0.068_{\pm 0.006}$ | $0.083_{\pm 0.006}$ | $0.104_{\pm 0.007}$ | $0.133_{\pm 0.007}$ |
| InfoGraph (Sun et al., 2019) | $0.072_{\pm 0.006}$ | $0.088_{\pm 0.006}$ | $0.106_{\pm 0.007}$ | $0.137_{\pm 0.007}$ |
| GraphLog (Xu et al., 2021) | $0.071_{\pm 0.005}$ | $0.089_{\pm 0.006}$ | $0.111_{\pm 0.006}$ | $0.138_{\pm 0.007}$ |
| GraphCL (You et al., 2020) | $0.075_{\pm 0.006}$ | $0.092_{\pm 0.006}$ | $0.112_{\pm 0.007}$ | $0.144_{\pm 0.008}$ |
| GraphMVP (Liu et al., 2022) | $0.078_{\pm 0.006}$ | $0.097_{\pm 0.007}$ | $0.118_{\pm 0.007}$ | $0.151_{\pm 0.008}$ |
| MolR (Wang et al., 2022) | $0.094_{\pm 0.006}$ | $0.114_{\pm 0.007}$ | $0.137_{\pm 0.007}$ | $0.170_{\pm 0.008}$ |
| ChemBerta-MTR (Ahmad et al., 2022) | $0.098_{\pm 0.006}$ | $0.121_{\pm 0.007}$ | $0.147_{\pm 0.008}$ | $0.183_{\pm 0.008}$ |
| GNN-MT | $0.112_{\pm 0.006}$ | $0.144_{\pm 0.007}$ | $0.177_{\pm 0.008}$ | $0.223_{\pm 0.009}$ |
| CLAMP (Seidl et al., 2023) | $0.202_{\pm 0.009}$ | $0.228_{\pm 0.009}$ | $0.248_{\pm 0.009}$ | $0.271_{\pm 0.009}$ |
| L-probe (ours) | $\mathbf{0.224}_{\pm 0.010}$ | $\mathbf{0.252}_{\pm 0.010}$ | $\mathbf{0.273}_{\pm 0.010}$ | $\mathbf{0.301}_{\pm 0.009}$ |

In the recent years, several methods were introduced proposing different ways to pre-train models in drug discovery. However, most of these models were not evaluated in a few-shot setting. We adapted several architectures on the tasks of FS-mol with the linear probe, and compare the results to the three methods already evaluated on this benchmark, that have been pretrained using the training set of FS-mol. The 14 backbones we selected for these experiments were pretrained on various datasets, such as PubChem (Wang et al., 2016), GEOM (Axelrod & Gómez-Bombarelli, 2022) and QMugs(Isert et al., 2021)), USPTO (Wang et al., 2022), and we used the weights shared publicly by the corresponding authors for each model.

We notice that none of the models enable competitive few-shot performances. This is most probably due to the fact that non of these approaches were using data originating from this benchmark in there pre-training phase. Among these models, we notice that ChemBerta-MTR (Ahmad et al., 2022) achieves the best performances, a model pretrained with a supervised multitask pretraining objective.

