# OpenReview forum: "A Strong Baseline for Molecular Few-Shot Learning"
_TMLR — Accepted by TMLR_

### Review · Reviewer_gzkF · 2024-10-20

**Summary Of Contributions:**

This article revisited the linear probing approach for drug discovery-related tasks, claiming that it is still a strong baseline and performs on par with the prevailing meta-learning-based techniques. In addition, the authors proposed a quadratic probing method, which substitutes the cosine-like similarity calculation in linear probing by the Mahalanobis distance. To resolve the issue of the Frobenius norm of the covariance matrices going to infinity when trained on gradient descent, the authors suggest block coordinate optimization, which breaks down the cross-entropy loss and updates the mean and covariance parameters iteratively. Experiments show that the probing methods have good performance on property prediction tasks as well as in the OOD settings.

**Audience:**

Yes

**Claims And Evidence:**

Yes

**Requested Changes:**

### Narration
- In "Introduction -> Meta-learning", the second paragraph basically claims "meta-learning is not applicable to black-box models as their weights are not accessible; but black-box models are valuable for drug discovery". There are redundancies around the first point in the narration, and the second point is not supported by citations.

- The "Introduction" section does not discuss enough why the black-box setting is essential for drug discovery. Relevant citations are missing.

- Figure 1: Critical legends and explanations are missing. I'm not sure how to interpret this figure, nor can I gain any insights about the difference between these two classifiers.

- It is advised to add equation (9) to the article body, probably around equation (2) or (3) for a clearer narration.

- It also would be good if the authors could add a figure or algorithm illustrating the training process of the proposed quadratic probing method.

- The authors should emphasize the difference between the meta-learning and fine-tuning approaches somewhere in the introduction.

### Supports
- In "Introduction -> Low data regime", it seems that none of the three cited papers (Schimunek et al. (2023); Chen et al. (2023); Altae-Tran et al. (2017)) supports the authors' claim "where meta-learning seems to appear as the default and best solution".

- In "Introduction -> Competitive fine-tuning methods", the authors only cited Stanley et al. (2021) to support their claim "fine-tuning approaches yielded subpar performances". But the majority work of that paper was conducted in August 2021, which was a "long" while ago in terms of the development of foundation models. A quick search on Google gives more advanced foundation models for drug discovery, such as [1], which may invalidate the authors' claim.

- Some related works such as [1] are missing. A recent survey [2] also mentioned some advanced techniques for drug discovery. If we widen the scope to molecular property prediction, then another survey [3] also showed a bunch of very related backbone models and their fine-tuning performance.

### Format
- It seems that the authors are using the citation format "(\citet{a}, \citet{b})". Instead, the authors should use "\citep{a,b}".
- In the LaTeX source code, quotation marks are "``" and "''" for left and right quotations.
- "Notations" should not be included in the "Related Work" section.
- The authors do not entirely follow the "Default Notations" defined in the TMLR & ICLR templates. Although it is not necessary, authors are suggested to follow those conventions to make readers' lives easier.
- The authors should not re-use the same notation for different concepts, e.g., $x_i$ are included in both support and query sets.

[1] Méndez-Lucio, Oscar, Christos Nicolaou, and Berton Earnshaw. "MolE: a molecular foundation model for drug discovery." arXiv preprint arXiv:2211.02657 (2022).
[2] Khan, Wasif, et al. "A Comprehensive Survey of Foundation Models in Medicine." arXiv preprint arXiv:2406.10729 (2024).
[3] Xia, Jun, et al. "A systematic survey of chemical pre-trained models." arXiv preprint arXiv:2210.16484 (2022).

**Strengths And Weaknesses:**

Although I know a bit about molecular property prediction and retrosynthesis, I am not entirely familiar with drug discovery tasks, meta-learning, or probing techniques. I tried to read the article as carefully as I could but I still might have missed or misunderstood some details.

## Strengths
- The proposed quadratic probing method seems solid and well-developed. Experiments also show its advantage over the linear probing method.
- The experiments look solid in terms of tasks. The discussion seems reasonable.
- The authors provide well-structured code which could facilitate further works.

## Weaknesses
- The narration of this article is not very fluent and could use some improvement. Please check the "Requested Changes" for details.
- Some of the claims are not well-supported or potentially outdated.
- In this article, the authors selected only a small scope of fine-tuning methods as the baseline. But there are some foundation models in the chemical domain released in recent years, such as those mentioned in [1, 2, 3]. The existence of such backbone models might impact the authors' claim "a strong baseline" in the title. Therefore it might be good for the authors to also use these models as baselines, or at least give a convincing discussion about whether these models are not applicable in the authors' setup.

## Questions
- Do the distributions of the support set and query set overlap? This information should be mentioned when these sets are introduced.
- For equation (1), can you give a reference to this type of linear probing classifier? I personally have never seen people put the bias term outside of the temperature scaling function so I want to verify whether this is correct.
- In section 3.3, "Instead of the cosine similarity used in the linear probe": I'm not sure whether the authors referred the "cosine similarity" to equation one. If so, I think it may not be accurate as temperature and bias terms are included.
- I'm also confused by the statement in the abstract "... and removing the need for specific episodic pre-training strategies". It seems that the proposed methods still utilize a pre-trained model as the backbone?

[1] Méndez-Lucio, Oscar, Christos Nicolaou, and Berton Earnshaw. "MolE: a molecular foundation model for drug discovery." arXiv preprint arXiv:2211.02657 (2022).
[2] Xia, Jun, et al. "A systematic survey of chemical pre-trained models." arXiv preprint arXiv:2210.16484 (2022).
[3] Zhou, Gengmo, et al. "Uni-mol: A universal 3d molecular representation learning framework." (2023).

---

> ### Author Response · Authors · 2024-10-26
> **Rebuttal by authors - Questions section**
>
> We thank Reviewer o2cb for their detailed review and the interesting questions they raised.
> ## Questions:
> **Do the distributions of the support set and query set overlap? This information should be mentioned when these sets are introduced.**
>
> In our paper, we define the support and query sets by partitioning the task, ensuring the two sets are distinct. Specifically, the support set $\mathcal{S}_t$ and the query set $\mathcal{Q}_t$​ do not overlap, i.e., $\mathcal{S}_t \cap \mathcal{Q}_t = \emptyset$. We will clarify this distinction explicitly in the notation section to avoid ambiguity.
>
> **For equation (1), can you give a reference to this type of linear probing classifier? I personally have never seen people put the bias term outside of the temperature scaling function so I want to verify whether this is correct.**
>
> The two formulations are indeed mathematically equivalent because they both represent linear transformations of the logits. In our paper, we define the logits of the linear probe as: $l_{i,k} = \tau\langle \mathbf{z}_i, \mathbf{w}_k \rangle + b_k$, where $\mathbf{w}_k$ and $b_k$ are optimized during the training.
>
>  This formulation is equivalent to $\tilde{l}_{i,k} = \tau\left(\langle \mathbf{z}_i, \mathbf{w}_k \rangle + \tilde{b}_k\right)$ by taking $\tilde{b}_k = b_k$ (since the bias is an optimized parameter of the probe, multiplications by constants do not affect the results at convergence).
>
>
>
> **In section 3.3, "Instead of the cosine similarity used in the linear probe": I'm not sure whether the authors referred the "cosine similarity" to equation one. If so, I think it may not be accurate as temperature and bias terms are included.***
>
> Following the previous answer, the temperature parameter is included in parameter $\Sigma_k^{-1}$ since it is just a linear transformation. We omitted the bias term for better clarity of the formulation, and because we found it to have a marginal impact on performances.
>
> **I'm also confused by the statement in the abstract "... and removing the need for specific episodic pre-training strategies". It seems that the proposed methods still utilize a pre-trained model as the backbone?**
>
> In this statement, we refer to the need for a pre-training strategy specific to few-shot learning. This statement might be unclear, and we will rephrase it to:
> ''and removing the need for episodic pre-training strategies specific to few-shot learning tasks.''

---

> > ### Author Response · Authors · 2024-10-26
> > **Rebuttal by authors - Requested Changes section**
> >
> > ## **Requested Changes:**
> > ### Narration
> > **First and second comments regarding the motivation of black-box models in drug discovery**
> >
> > We acknowledge the redundancies and the need for stronger support in this section. To address this, we will add citations [3, 4, 5] to substantiate the claim that "data is highly sensitive, and sharing a dataset or a pre-trained model on a private dataset is not always possible." In such cases, black-box models are particularly valuable because they enable the use of pre-trained models without exposing their weights.
> >
> > [3] Wouter Heyndrickx, Lewis Mervin, et al. MELLODDY: Cross-pharma Federated Learning at Unprecedented Scale Unlocks Benefits in QSAR without Compromising Proprietary Information. Journal of Chemical Information and Modeling, April 2024.
> >
> > [4] Balazs Pejo, Mina Remeli et al. Collaborative Drug Discovery: Inference-level Data Protection Perspective, June 2022. arXiv:2205.06506
> >
> > [5] Aljoša Smajić, Melanie Grandits, and Gerhard F. Ecker. Privacy-preserving techniques for decentralized and secure machine learning in drug discovery. Drug Discovery Today, 28(12):103820, December 2023.
> >
> > **Comment on Figure 1**
> >
> > We will revise the figure's legend. The figure illustrates the values of the logits produced by two different classifiers: a linear probe and a quadratic probe based on the Mahalanobis distance, in a two-dimensional feature space with two classes.
> >
> > **It is advised to add equation (9) to the article body, probably around equation (2) or (3) for a clearer narration.**
> >
> > We will add this equation to the article body. Thanks for the suggestion!
> >
> > **It also would be good if the authors could add a figure or algorithm illustrating the training process of the proposed quadratic probing method.**
> >
> > We will add the algorithm describing the training process of the quadratic probe to the article body, which will be available in the first revision of the paper once all three reviews will be published.
> >
> > **The authors should emphasize the difference between the meta-learning and fine-tuning approaches somewhere in the introduction.**
> >
> > To clarify the differences between both approaches, we will modify the first part of the fine-tuning methods' paragraph as follows:
> >
> > "Fine-tuning methods involve using a model pre-trained in a standard supervised setting and then fine-tuning it for few-shot classification. Unlike meta-learning, fine-tuning approaches utilize pre-trained models that can be applied to other tasks in standard machine learning, beyond few-shot classification."
> >
> > ## Supports
> > **In "Introduction -> Low data regime", it seems that none of the three cited papers (Schimunek et al. (2023); Chen et al. (2023); Altae-Tran et al. (2017)) supports the authors' claim "where meta-learning seems to appear as the default and best solution".**
> >
> > The cited papers (Schimunek et al., 2023; Chen et al., 2023; Altae-Tran et al., 2017) evaluate meta-learning approaches for few-shot classification on the FS-MOL benchmark, displaying highly competitive results. In all three cases, fine-tuning baselines were found to be less effective, as they did not perform competitively on the evaluated tasks. We believe this supports the statement in our paper by illustrating the observed trend where meta-learning emerges as a more suitable approach for few-shot learning in these scenarios.
> >
> > **Third and Fourth comments on molecular foundational models.**
> >
> > In the introduction, we address the line of research on molecular foundation models. To the best of our knowledge, only CLAMP (ICML 2023) was evaluated in the few-shot setting (which we included in our baselines), achieving satisfactory results.
> >
> > We will extend this discussion by including [1,2,3] to the discussion, however, it is important to note that none of the models discussed in these works have been evaluated in a few-shot learning context.
> >
> > We evaluated the 15 backbones mentioned in [3], fine-tuning them with a linear probe. The table available in the following comment shows that none of these models achieve performances similar to CLAMP.
> > This disparity can be attributed to the fact that none of these models were specifically trained for Quantitative Structure-Activity Relationship (QSAR) modeling or on QSAR datasets. In contrast, our backbone, GNN-MT, and CLAMP were trained using the FS-MOL dataset, which likely contributed to their superior performance in this setting.
> >
> > *The weights of the MolE model [1] are not publicly available, so we could not evaluate this architecture*
> > ## Format
> > Thank you for your observations regarding the formatting. We will implement the necessary adjustments as highlighted in your review.
> > We understand your concern about including "Notations" in the "Related Work" section. However, some of the notations we introduce are used in that section, which justifies their placement in the paper. Moving this section could make it harder for readers to understand the context. Thank you for your understanding.

---

> > > ### Author Response · Authors · 2024-10-26
> > > **Evaluation of Molecular Representation Learning / Foundational models in a Few-shot setting**
> > >
> > > Performances of various backbones to enable molecular few-shot classification.
> > >
> > > | model           | $\mid S \mid$ = 16    | $\mid S \mid$ = 32    | $\mid S \mid$ = 64    | $\mid S \mid$ = 128   |
> > > | :-------------- | :-------------------- | :-------------------- | :-------------------- | :-------------------- |
> > > | MolBert         | 0.045 $\pm$ 0.005     | 0.058 $\pm$ 0.005     | 0.074 $\pm$ 0.006     | 0.105 $\pm$ 0.007     |
> > > | AttributeMask   | 0.048 $\pm$ 0.005     | 0.060 $\pm$ 0.005     | 0.077 $\pm$ 0.006     | 0.110 $\pm$ 0.007     |
> > > | ChemBertMLM-10M | 0.049 $\pm$ 0.005     | 0.063 $\pm$ 0.006     | 0.080 $\pm$ 0.006     | 0.113 $\pm$ 0.007     |
> > > | ThreeDInfomax   | 0.050 $\pm$ 0.004     | 0.067 $\pm$ 0.005     | 0.081 $\pm$ 0.006     | 0.108 $\pm$ 0.006     |
> > > | ContextPred     | 0.052 $\pm$ 0.005     | 0.067 $\pm$ 0.005     | 0.082 $\pm$ 0.006     | 0.113 $\pm$ 0.007     |
> > > | GPT-GNN         | 0.056 $\pm$ 0.005     | 0.070 $\pm$ 0.005     | 0.085 $\pm$ 0.006     | 0.114 $\pm$ 0.006     |
> > > | MolFormer       | 0.058 $\pm$ 0.006     | 0.075 $\pm$ 0.006     | 0.092 $\pm$ 0.007     | 0.125 $\pm$ 0.008     |
> > > | MAT             | 0.052 $\pm$ 0.005     | 0.069 $\pm$ 0.006     | 0.092 $\pm$ 0.007     | 0.136 $\pm$ 0.009     |
> > > | GROVER          | 0.068 $\pm$ 0.006     | 0.083 $\pm$ 0.006     | 0.104 $\pm$ 0.007     | 0.133 $\pm$ 0.007     |
> > > | InfoGraph       | 0.072 $\pm$ 0.006     | 0.088 $\pm$ 0.006     | 0.106 $\pm$ 0.007     | 0.137 $\pm$ 0.007     |
> > > | GraphLog        | 0.071 $\pm$ 0.005     | 0.089 $\pm$ 0.006     | 0.111 $\pm$ 0.006     | 0.138 $\pm$ 0.007     |
> > > | GraphCL         | 0.075 $\pm$ 0.006     | 0.092 $\pm$ 0.006     | 0.112 $\pm$ 0.007     | 0.144 $\pm$ 0.008     |
> > > | GraphMVP        | 0.078 $\pm$ 0.006     | 0.097 $\pm$ 0.007     | 0.118 $\pm$ 0.007     | 0.151 $\pm$ 0.008     |
> > > | MolR_tag        | 0.094 $\pm$ 0.006     | 0.114 $\pm$ 0.007     | 0.137 $\pm$ 0.007     | 0.170 $\pm$ 0.008     |
> > > | ChemBertMTR-77M | 0.098 $\pm$ 0.006     | 0.121 $\pm$ 0.007     | 0.147 $\pm$ 0.008     | 0.183 $\pm$ 0.008     |
> > > | GNN-MT          | 0.112 $\pm$ 0.006     | 0.144 $\pm$ 0.007     | 0.177 $\pm$ 0.008     | 0.223 $\pm$ 0.009     |
> > > | CLAMP           | 0.201 $\pm$ 0.009     | 0.226 $\pm$ 0.009     | 0.246 $\pm$ 0.009     | 0.269 $\pm$ 0.009     |
> > > | L-probe (ours)  | **0.224 $\pm$ 0.010** | **0.252 $\pm$ 0.010** | **0.273 $\pm$ 0.010** | **0.301 $\pm$ 0.009** |

---

### Review · Reviewer_ANKt · 2024-10-23

**Summary Of Contributions:**

The authors propose a simple yet effective baseline for molecular few-shot learning. The approach is very simple; it performs linear/quadratic probing of the pre-trained model. Experimental results show that the proposed approach achieves nearly state-of-the-art performance on molecular few-shot learning.

**Audience:**

Yes

**Claims And Evidence:**

Yes

**Requested Changes:**

Please see Weaknesses.

**Strengths And Weaknesses:**

Strengths:
1. The problem of interest, molecular few-shot learning, is an important topic in practical applications, e.g., drug discovery.

2. The method achieves good (nearly state-of-the-art) performance on various molecular few-shot learning tasks.

3. The authors performed useful analysis on the effectiveness of the proposed method.

Weakness:

1. My biggest question is "how" linear/quadratic probing is different from GNN-MT in the original FS-Mol paper. Linear/quadratic probing (in this paper) seems very similar to GNN-MT; the pre-training objective is the same, and the only difference is the fine-tuning scheme (this paper freezes the backbone while GNN-MT fine-tunes the backbone). However, the reported performance is dramatically different in Table 1. Could the authors provide an analysis on this phenomenon?

2. Compared to linear probing, the improvement of quadratic probing is marginal. Also, it seems that quadratic probing requires more complexity than linear probing (due to block-coordinate descent and the estimation of covariance matrix's inverse).

3. Empirical support of the claim on quadratic probing is weak. The authors claimed that linear probing yields "degenerate solutions", but I could not find the definition of degenerate solutions and how quadratic probing can resolve it. Also, the authors claimed that the proposed quadratic probing can prevent the divergence of the eigenvalues of M_k, but there is no empirical support.

4. This paper is poorly formatted, e.g., "))" at the end of the reference and inconsistent "." at the end of the sub-subsection.

---

> ### Author Response · Authors · 2024-10-26
> **Rebuttal by Authors**
>
> We thank reviewer ANKt for their review of our work.
>
> ## Weaknesses
>
> **1) My biggest question is "how" linear/quadratic probing is different from GNN-MT in the original FS-Mol paper. Linear/quadratic probing (in this paper) seems very similar to GNN-MT; the pre-training objective is the same, and the only difference is the fine-tuning scheme (this paper freezes the backbone while GNN-MT fine-tunes the backbone). However, the reported performance is dramatically different in Table 1. Could the authors provide an analysis on this phenomenon?**
>
> The proposed method does not focus on the pretraining stage, but only on the few-shot adaptation process. The difference between GNN-MT and our method indeed resides in the adaptation phase. While GNN-MT fine-tunes the whole backbone and adds a task-specific 2-layer perceptron, we freeze the backbone and train only the probing classifiers.
> GNN-MT thereby fine-tunes orders of magnitudes more parameters than our method. Therefore, it is indeed expected to perform poorly when the training is done in a very low-data regime (i.e., on 16 samples). In fact, in other, completely different application domains, such as computer vision, it is well-known that full fine-tuning performs poorly in few-shot regimes. Surprisingly, this fact has been overlooked in the few-shot drug discovery literature and probing baselines (with frozen backbones) are not evaluated.
>
> **2) Compared to linear probing, the improvement of quadratic probing is marginal. Also, it seems that quadratic probing requires more complexity than linear probing (due to block-coordinate descent and the estimation of covariance matrix's inverse).**
>
> While this observation is true on the FS-mol benchmark (i.e., with small task sizes), the difference between the two methods increases with the support set's size, as one would expect. Please notice that, in the library screening experiments, when the support set size exceeds 32, the linear probe is outperformed by the other baselines. In contrast, the quadratic probe remains highly competitive, and is outperformed only by the similarity search approach.
>
> In terms of complexity, evaluating a task with both probes takes on average less than 2 seconds for both methods (1.43s for the linear probe and 1.49s with the quadratic one).
>
> **3) Empirical support of the claim on quadratic probing is weak. The authors claimed that linear probing yields "degenerate solutions", but I could not find the definition of degenerate solutions and how quadratic probing can resolve it. Also, the authors claimed that the proposed quadratic probing can prevent the divergence of the eigenvalues of M_k, but there is no empirical support.**
>
> In fact, we do not claim that linear probing yields degenerate solutions. The degenerate solutions only appear in the quadratic probing, when directly optimizing the cross-entropy loss with respect to matrices $M_k$. Proposition 1 shows that such a direct optimization achieves a perfect loss value (a cross-entropy of 0) when the largest eigenvalue of $M_k$ diverges (i.e., goes to infinity, which is referred to as a degenerate solution). We will clarify this definition of a degenerate solution. We provide empirical support of this claim in the following figure showing the evolution of the value of $M_k$'s largest eigenvalue by directly optimizing it (Free-opt), and using the quadratic probe (q-probe).
>
> The Figure can be accessed in the following anonymous repository: https://anonymous.4open.science/r/tmlr_rebutt-D843/README.md
>
> **Format**
>
> Thank you for your observations regarding the formatting of our paper, and we will correct the citations and subsection's name.

---

### Review · Reviewer_WriN · 2024-11-02

**Summary Of Contributions:**

In this work, the authors demonstrate that a simple pre-training/fine-tuning strategy with probing leads to competitive results with far more involved meta-learning approaches. In addition, probing has the benefit of being applicable in black-box settings where the pre-trained model is only accessible through an API. The authors further develop an alternative to the standard linear probing, called quadratic probing, a generalization of linear probing that is more expressive in separating data and expected to perform better. As the authors show, a downside of quadratic probing is that it cannot be successfully optimized with gradient descent. To still make use of its benefits, the authors develop an approach with block coordinate optimization. Empirically, the block coordinate optimization leads to a more robust training behavior and to a competitive performance of quadratic probing when compared to state-of-the-art meta-learning approaches. In addition, both linear and quadratic probing perform strongly in a domain shift setting with unbalanced data.

**Audience:**

Yes

**Broader Impact Concerns:**

No concerns.

**Claims And Evidence:**

Yes

**Requested Changes:**

**Important requested changes:**
* Adding a more nuanced discussion on the benefits of the quadratic probe over the linear probe, considering that the linear probe performs very closely to the quadratic probe but is conceptually simpler and easier to optimize with standard procedures (e.g., gradient descent).

**Minor remarks that would strengthen the work but are not critical to my recommendation:**
* The authors show a clear understanding of drug discovery and the applications of machine learning therein. Since many ML researchers may be interested in contributing to this field but lack the necessary background to understand the broader task that drug discovery and ML-aided drug discovery tries to solve, the authors could add a background section (perhaps to the appendix), outlining the overall drug discovery process and indicate where ML approaches could make useful impact. In addition, such a section would provide space for defining some terminology that is common in drug discovery but not known to the broader ML community. I think this would greatly enhance the accessibility of the author's research.

**Strengths And Weaknesses:**

**Strengths:**
* The motivation is clear and the topic important and relevant.

* The work takes the black-box setting into account without access to model weights in downstream applications. In the context of large pre-trained models and the costs of serving such models, considering this setting is sensible.

* The authors do not only include state-of-the-art baselines but further include classical baselines such as kNNs, random forests and simple GNN baseline. When working on tasks with very few data points, adding such baselines is a very good practice. In my opinion, this strengthens the impact of this work.

**Weaknesses:**
* A large portion of the paper is concerned with the quadratic probe. Using this approach appears much more involved as the linear probe. In particular, the fact that optimizing the quadratic probe with gradient descent leads to degenerate solutions indicates to me that this approach is less robust in real-world scenarios. And yet, the performance of the quadratic probe does not seem to lead to significant improvements over the linear probe. In Table 1 and Table 2 (QSAR datasets), all linear probe results are within one standard deviation of the quadratic probe. In light of these results, the authors should add more discussion on the benefits of the quadratic probe over the linear probe. I want to mention, however, that this does not invalidate the claim made by the authors that the quadratic probe performs better than the linear probe. I do recognize that the quadratic probe consistently leads to better results. I merely wonder whether the additional effort in optimizing the quadratic probe is justified by the improvements. There may not be a perfect answer to this question, but I do recommend to add a more nuanced discussion on this limitation.

---

> ### Author Response · Authors · 2024-11-08
> **Rebuttal by authors**
>
> We thank reviewer WriN for their thorough review of our work.
>
> ## Weakness and Requested Changes:
>
> **Comparison of Performances between the Linear and Quadratic probes**
>
> Thank you for your valuable feedback. We agree that each probe has its strengths and limitations, and we acknowledge the need for a more nuanced discussion of the quadratic probe’s benefits and the context in which it provides value over the linear probe.
>
> Indeed, the linear probe achieves similar results on the FS-mol benchmark, especially in low-data regimes, and it provides a straightforward baseline with lighter computational requirements.
>
> In contrast, the quadratic probe requires an additional step, the covariance matrix estimation.
> While it is more complex than the linear probe, it remains relatively simple, especially compared to pre-existing meta-learning approaches.
> To clarify its implementation, we provide an algorithm describing its training process in the appendix of the revised version of the paper.
> Furthermore, the quadratic probe consistently shows performance improvements, which increase with the support set size, and demonstrates enhanced robustness on specific data distributions, such as library screening tasks. In those tasks, we observe that the linear probe's performance degrades significantly as the labeled data size exceeds 32 points (cf. Table 2 and Figure 4).
>
> In terms of runtime, on an A6000 GPU, both methods are efficient, running within 2 seconds per task on average (1.43s for the linear probe and 1.49s for the quadratic one). This marginal difference in runtime highlights that the additional complexity of the quadratic probe does not substantially impact the computational costs.
>
> Ultimately, the choice between linear and quadratic probes will depend on dataset characteristics and application-specific requirements, with both methods presenting valuable options. We will add a more nuanced discussion of these strengths and limitations in the results section on FS-mol and in the conclusion (see modifications in the revised version of the paper).
>
>
> **Background section**
>
> We agree adding a section in the appendix providing some background information on drug discovery might increase the reach of this article. We are currently working on this section and will add it to the appendix of our work.

---

### Decision · Action_Editor_ftTa · 2025-01-06

**Recommendation:** Accept as is

**Comment:**

The reviewers unanimously recommend acceptance, particularly in light of the black-box setting where probing is a natural fit. For the camera ready version, the reviewers are encouraged to incorporate any relevant feedback from the reviewers in order to improve the paper.

**Audience:**

The primary audience of this paper are researchers and practitioners interested in few-shot learning for drug discovery. Those involved in other applications of few-shot learning may also be interested due to the black-box setting, which is increasingly relevant as models are often only made available through API calls.

**Claims And Evidence:**

This paper investigates the problem of few-shot learning for drug discovery, in which the goal is to discover chemical compounds with specific desired properties. Few-shot learning is important in drug discovery due to the expense of obtaining data from wet lab experiments. An additional relevant aspect of this setting is that the underlying pre-trained model is often a black box due to data-sharing or computational concerns. In order to address these challenges, this paper proposes a quadratic probing loss and corresponding optimizer. Performance is found to be competitive with state-of-the-art meta-learning methods while still being amenable to the black box setting.

There are three main claims introduced in this paper. The first is the contribution of results from a standard linear probe, which is often overlooked in the drug discovery setting. The second is the proposal of a more expressive quadratic probe and corresponding block-coordinate descent optimizer that achieves competitive performance. The third is the introduction of robustness benchmarks that evaluate performance under distribution shift in the presence of imbalanced labels. Reviewers found that each of these claims are sufficiently supported by evidence. Probing results are compared to a range of meta-learning methods in Section 4.1 and the robustness benchmarks are presented in Section 4.3.